# un²CLIP: Improving CLIP's Visual Detail Capturing Ability via Inverting unCLIP

**Yinqi Li**[1,2], **Jiahe Zhao**[1,2], **Hong Chang**[1,2✉], **Ruibing Hou**[1], **Shiguang Shan**[1,2], **Xilin Chen**[1,2]

[1]State Key Laboratory of AI Safety, Institute of Computing Technology, CAS, China
[2]University of Chinese Academy of Sciences (CAS), China
yinqi.li@vipl.ict.ac.cn, {zhaojiahe22s, changhong, houruibing, sgshan, xlchen}@ict.ac.cn

## Abstract

Contrastive Language-Image Pre-training (CLIP) has become a foundation model and has been applied to various vision and multimodal tasks. However, recent works indicate that CLIP falls short in distinguishing detailed differences in images and shows suboptimal performance on dense-prediction and vision-centric multimodal tasks. Therefore, this work focuses on improving existing CLIP models, aiming to capture as many visual details in images as possible. We find that a specific type of generative models, unCLIP, provides a suitable framework for achieving our goal. Specifically, unCLIP trains an image generator conditioned on the CLIP image embedding. In other words, it inverts the CLIP image encoder. Compared to discriminative models like CLIP, generative models are better at capturing image details because they are trained to learn the data distribution of images. Additionally, the conditional input space of unCLIP aligns with CLIP's original image-text embedding space. Therefore, we propose to invert unCLIP (dubbed un²CLIP) to improve the CLIP model. In this way, the improved image encoder can gain unCLIP's visual detail capturing ability while preserving its alignment with the original text encoder simultaneously. We evaluate our improved CLIP across various tasks to which CLIP has been applied, including the challenging MMVP-VLM benchmark, the dense-prediction open-vocabulary segmentation task, and multimodal large language model tasks. Experiments show that un²CLIP significantly improves the original CLIP and previous CLIP improvement methods. Code and models are available at https://github.com/LiYinqi/un2CLIP.

## 1 Introduction

Contrastive Language-Image Pre-training (CLIP) models trained on web-scale datasets have shown great success in learning transferable representations for image classification [1]. Since their release, they have been widely adopted for many vision and multimodal tasks, such as open-vocabulary segmentation [2, 3, 4] and multimodal large language model (MLLM) tasks [5, 6, 7]. These extensions can be attributed to the learned web-scale knowledge and the vision-language alignment property of CLIP models. However, CLIP cannot always perform well on these extended, finer-grained tasks that require more detailed image understanding, as observed in dense vision tasks [2, 8] and MLLM literature [9]. This phenomenon may be due to CLIP's training data and objective, where the global-level image-text contrastive learning makes the image encoder poor at capturing visual details.

To address this problem, some works modify the network architecture at inference time to make the encoder gather less global information [2, 3, 4], some other works deploy additional visual self-supervised encoders in dense-vision tasks [10, 11] and MLLM frameworks [9, 12, 13, 14]. Although effective, these methods do not fundamentally address the problem that CLIP falls short of capturing visual details. To build finer-grained CLIP models, existing works mainly focus on training new

39th Conference on Neural Information Processing Systems (NeurIPS 2025).

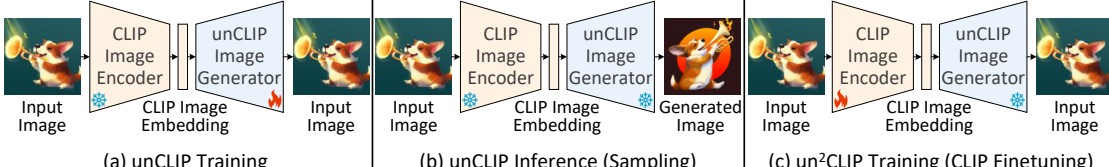

| (a) unCLIP Training | (b) unCLIP Inference (Sampling) | (c) un²CLIP Training (CLIP Finetuning) |

Figure 1: Comparison of unCLIP [20] and un²CLIP pipelines. (a) (b) unCLIP provides an encoding-decoding tool for observing which features of the image are disregarded by CLIP (more examples are shown in Figure 3 of [20]). (c) Our un²CLIP further leverages this framework to improve CLIP, aiming to recapture the disregarded features. The unCLIP model contains a text to image-embedding "prior" module for supporting the text-to-image pipeline, which is not used in our work.

CLIP variants with more detailed vision-text supervisions such as region-text alignment [8, 15, 16, 17, 18, 19]. However, collecting high-quality region-text paired datasets in real world is more difficult than acquiring original CLIP's image-text pairs, because the former is limited in amount at web and typically requires human annotation. Besides, re-training CLIP models is costly that we would like to avoid. Therefore, in this work, we focus on improving existing CLIP models from the vision-centric perspective using image data only.

Specifically, we find that a specific type of image generation models, unCLIP [20], provides a suitable framework for achieving our goal. To start with, firstly, unCLIP offers a way to qualitatively visualize which features are disregarded by the pretrained CLIP image encoder. As illustrated in Figure 1(a), unCLIP trains an image generator that inverts the CLIP image encoder, taking the CLIP image embedding as input to generate images. After training, the encoding-decoding pipeline provides us a tool to observe which features of the image are not captured by the encoder, as shown in Figure 1(b).

Furthermore, this encoding-decoding pipeline not only serves as a tool to observe CLIP's failures, *but also* offers a proper framework to improve CLIP in a vision-centric manner. To be specific, building a finer-gained CLIP requires enhancing the visual detail capturing ability of its image encoder while preserving the image-text alignment property at the same time. We note that the unCLIP framework is suitable for achieving this goal because (1) unCLIP is a generative model learning the underlying distribution of image data, which enhances its capability in capturing the complexities and variations in images; (2) unCLIP takes the image embeddings from the output layer of the pretrained CLIP as its conditional input, which are aligned with their corresponding CLIP text embeddings.

Based on these two properties, we propose to finetune the CLIP encoder in an unCLIP generator inversion way, as illustrated in Figure 1(c), thereby transferring the generator's rich visual knowledge into the encoder and leveraging the remarkable language-alignment property of the unCLIP generator's input space. We name our method un²CLIP, since it inverts the unCLIP image generator. Compared to prior work [21], which improves the CLIP image encoder using a pretrained text-to-image generative model that has a mismatched input space with CLIP image embeddings, our un²CLIP is designed within the CLIP-embedding-aligned framework, thereby facilitating more effective finetuning.

Our contributions can be summarized as follows:

- We find that unCLIP provides a suitable framework for improving the CLIP image encoder's ability to capture visual details while maintaining its alignment with the language encoding space.
- Based on the above finding, we propose a pure image-based method un²CLIP to improve CLIP, which finetunes the CLIP image encoder by inverting unCLIP.
- Our experiments show that un²CLIP significantly improves CLIP and outperforms prior methods on various tasks and benchmarks, including the MMVP-VLM benchmark consisting of CLIP-blind pairs [9], the dense vision task of open-vocabulary segmentation, and vision-centric MLLM tasks.

## 2  Related Works

**CLIP and Downstream Applications.**  CLIP trains a pair of image and text encoders using a contrastive loss on web-collected image-text pairs, which learns transferable representations and shows remarkable zero-shot classification performance [1]. Due to its large-scale pretraining and the vision-language alignment property, CLIP has become a foundation model, widely applied across a variety of

vision and multimodal tasks, including open-vocabulary semantic segmentation [2, 3, 4, 22, 23, 24], objet detection [8, 15, 17, 25, 26], text-to-image generation [20, 27, 28, 29], and large-language-model-based visual question answering [30, 5, 6, 7]. However, recent studies have shown out that CLIP underperforms on tasks that require fine-grained visual understanding. Tong et al. [9] find that CLIP struggles to distinguish certain visual patterns, which also impairs the performance of MLLMs built on top of CLIP. Additionally, [2, 8] claim that CLIP, being trained with the image-level objective, tends to capture global semantics while neglecting local details, limiting its effectiveness as a backbone for dense prediction tasks.

**Vision-centric CLIP Improvements.** To improve CLIP's ability to perform fine-grained visual tasks, existing efforts can be broadly categorized into two groups. The first group targets specific downstream applications. For instance, some works [2, 3, 4] modify CLIP's image encoder architecture at inference time to make it gather less global-level information, thereby improving performance on the pixel-level dense prediction task. Targeted at vision-centric MLLM tasks, works [9, 12, 13, 14] incorporate additional visual self-supervised encoders, and [31] finetunes the CLIP encoder with visual supervision within the MLLM framework, to address CLIP's shortcoming in learning detailed image representations. Although effective, these approaches are downstream-task-specific and do not fundamentally resolve CLIP's intrinsic limitation in modeling detailed visual representations.

On the other hand, another line of work seeks to improve CLIP at the upstream stage by modifying its training data and objectives to produce finer-grained representations. Specifically, several approaches [8, 15, 16, 17, 18, 19] perform CLIP-like pretraining but align image regions with textual descriptions, enabling the model to learn more detailed, region-level visual representations. However, collecting high-quality, large-scale region-text pairs is more challenging than acquiring image-text pairs, as the former typically requires human annotation, while the latter can be easily sourced via web scraping. Our work also aims to fundamentally address CLIP's limitation in capturing visual details at the upstream level. In contrast to region-level approaches, we rely solely on image data and leverage generative models to enhance visual detail capturing. The most relevant prior work is DIVA [21], which uses a pretrained text-to-image generative model whose input mismatches CLIP's output to improve CLIP. In comparison, our framework uses a generative model that operates in the same representation space as CLIP, enabling a more effective and seamless enhancement process.

**Generative Models for Representation Learning.** Generative models such as diffusion models [32, 33] trained on large-scale datasets have shown remarkable progress in generating photorealistic images [34, 27, 28, 29]. The remarkable generation ability implies that they have accurately modeled the underlying data distribution and learned effective representations of images. Based on this motivation, [35, 36, 37, 38, 39] leverage the pretrained generative models as the backbone for visual perception, and show impressive performance on solving dense vision tasks such as semantic segmentation and depth estimation. This work also leverages the rich representations in generative models for visual perception, but we transfer them into an existing discriminative model - CLIP.

## 3  Method

We first introduce some preliminaries of generative models and unCLIP in Section 3.1, and then describe our un$^2$CLIP method in Section 3.2.

### 3.1  Preliminaries

**Generative Models.** Generative models, or image generation models in specific, are trained to learn the data distribution of images usually by maximizing the log-likelihood $\mathbb{E}_{\mathbf{x}} \log p_G(\mathbf{x})$, or $\mathbb{E}_{\mathbf{x}} \log p_G(\mathbf{x}|\mathbf{c})$ for conditional generative models, where $G$ denotes the generator, $\mathbf{x}$ stands for images, and $\mathbf{c}$ is the conditional input such as class and text.

Diffusion models [32, 33] are one of the popular types of generative models recently. They are trained to reverse a forward diffusion process, where noises $\epsilon \sim \mathcal{N}(\mathbf{0}, \mathbf{I})$ are gradually added to the clean image $\mathbf{x}$ over a number of timesteps $t = 1, \cdots, T$. For the reverse process, a denoising network $\epsilon_G(\mathbf{x}_t, t)$ is trained to estimate the added noise $\epsilon$ at each timestep, where $\mathbf{x}_t$ is the noisy image at $t$. Finally, the log-likelihood maximization objective is implemented by optimizing a variational bound of it:

$$\max_G \mathbb{E}_{\mathbf{x}} \left[ \log p_G(\mathbf{x}) \right] \approx \min_G \mathbb{E}_{\mathbf{x}, \epsilon, t} \left[ ||\epsilon - \epsilon_G(\mathbf{x}_t, t)||_2^2 \right]. \tag{1}$$

**unCLIP: CLIP-embedding-conditioned Generative Model.** Given a pretrained CLIP [1] image encoder $E$, unCLIP [20] aims to train a decoder that can invert the encoder $E$. The decoder is designed to be non-deterministic, meaning it can produce multiple images for a given CLIP image embedding. This allows for a more comprehensive examination of CLIP's disregarded features. Specifically, the decoder is implemented as a probabilistic image generator $G$ conditioned on the image embeddings from the output layer of CLIP image encoder, as illustrated in Figure 1(a). In [20], the authors employ diffusion model as the generator, using the following training objective:

$$\max_G \mathbb{E}_{\mathbf{x}} \left[ \log p_G(\mathbf{x}|E(\mathbf{x})) \right] \approx \min_G \mathbb{E}_{\mathbf{x},\epsilon,t} \left[ ||\epsilon - \epsilon_G(\mathbf{x}_t, t, E(\mathbf{x}))||_2^2 \right]. \tag{2}$$

## 3.2  un$^2$CLIP: Improving CLIP via Inverting unCLIP

CLIP models have been widely adopted in many vision and multimodal understanding tasks [2, 5, 6]. However, recent studies have revealed that CLIP falls short in distinguishing certain visual-detail related patterns [9] and exhibits suboptimal performance in solving dense vision tasks [2, 8]. This section aims to develop an approach to alleviate this problem.

Since original CLIP models are trained on global-level image-text pairs, developing a new CLIP variant that focuses on visual details from scratch poses challenges both in designing suitable training objectives and in collecting training data pairs, where high-quality, open-world region-text or pixel-text pairs would be difficult to collect. Moreover, re-training CLIP is costly. Therefore, we would like to improve existing pretrained CLIP models from the vision-centric perspective. To be specific, given a pretrained CLIP model, we aim to improve its image encoder to capture as many visual details in images as possible while maintaining its language-alignment property simultaneously.

**Goal of Our Problem.** Formally, the goal is to maximize the mutual information between an input image $\mathbf{x}$ and its embedding $E(\mathbf{x})$ produced by the CLIP image encoder, subject to a language-alignment constraint:

$$\max_E I(\mathbf{x}; E(\mathbf{x})), \text{ s.t. } d(E(\mathbf{x}), \mathbf{y}) \to 0, \tag{3}$$

where $\mathbf{y}$ denotes the semantic aligned text embedding of $\mathbf{x}$, produced by the CLIP text encoder, and $d$ stands for distance.

**Backend of Our Framework.** Note that the mutual information in Eq. (3) can be expressed as $I(\mathbf{x}; E(\mathbf{x})) = H(\mathbf{x}) - H(\mathbf{x}|E(\mathbf{x}))$ in terms of entropy $H$. Therefore, $\max_E I(\mathbf{x}; E(\mathbf{x}))$ in Eq. (3) equals to

$$\min_E H(\mathbf{x}|E(\mathbf{x})) = \min_E \mathbb{E}_{\mathbf{x}} \left[ -\log p(\mathbf{x}|E(\mathbf{x})) \right] = \max_E \mathbb{E}_{\mathbf{x}} \left[ \log p(\mathbf{x}|E(\mathbf{x})) \right], \tag{4}$$

where the first equation is according to the definition of the conditional entropy.

By comparing Eq. (4) with Eq. (2), we observe that the pretrained unCLIP model, which takes the CLIP image embedding $E(\mathbf{x})$ as input to generate images $\mathbf{x}$, provides a suitable probability model $p_G(\mathbf{x}|E(\mathbf{x}))$ for estimating Eq. (4). This motivates us to adopt the pretrained unCLIP model $G(E(\cdot))$ as the backend of our framework, enabling us to improve the front-end CLIP image encoder $E(\cdot)$, as illustrated in Figure 1(c).

**Reducing Language-shift During Finetuning.** The other factor remaining in our objective (Eq. (3)) is to make embeddings produced by the finetuned image encoder have correct semantics that can be interpreted by the original CLIP text encoder. Since our framework relies on image data only, maintaining this language-alignment property poses a nontrivial challenge. For addressing this challenge, we note that the input space of the unCLIP generator $G$ lies within the CLIP image-text embedding space. Therefore, for reducing the potential language-shift during the image-encoder finetuning, we *freeze* the parameters of $G$, thereby encouraging the optimized embedding $E(\mathbf{x})$, when fed to $G$, staying close to $G$'s original input domain.

**un$^2$CLIP.** Taking these together, our CLIP finetuning objective is

$$\max_E \mathbb{E}_{\mathbf{x}} \left[ \log p_G(\mathbf{x}|E(\mathbf{x})) \right] \approx \min_E \mathbb{E}_{\mathbf{x},\epsilon,t} \left[ ||\epsilon - \epsilon_G(\mathbf{x}_t, t, E(\mathbf{x}))||_2^2 \right]. \tag{5}$$

The training objective can be interpreted as inverting the unCLIP image generator $G$, which possesses strong visual detail representations and has an input space aligned with the embedding space of CLIP. By optimizing the CLIP image encoder $E$ in this unCLIP-inversion (dubbed un$^2$CLIP) framework, we effectively enhance $E$'s ability to capture fine-grained visual information. In practice, when finetuning

$E$ using Eq. (5), we inherit the training configuration of the unCLIP diffusion model, such as the timestep and noise schedule. This is like a replay of the training procedure of unCLIP $G$, but here we freeze the trained $G$ and update $E$.

**Comparison to the Prior Work DIVA [21].** DIVA is a related approach that also leverages pretrained generative models to enhance CLIP. However, its architecture and optimization differ fundamentally from ours. DIVA deploys a text-to-image generative model [28] behind the CLIP image encoder. Notably, the input space of such generators differs from the output space of the CLIP image encoder, both in embedding dimensionality and, more importantly, in semantic representation. To bridge this gap, DIVA inserts a trainable projection layer $P$ between the encoder and the generator. Its resulting training objective can be written as $\max_{E,P} \mathbb{E}_{\mathbf{x}} \left[\log p_G(\mathbf{x}|P(E(\mathbf{x})))\right]$, which deviates the derived goal in Eq. (4), where $P$ would take away part of the knowledge learned from $G$. Therefore, using a generator that is misaligned with the CLIP embedding space may be suboptimal for improving the CLIP image encoder.

## 4 Experiments

### 4.1 Experimental Setup

**Pretrained CLIP and unCLIP Models.** Since Ramesh et al. [20] do not provide official unCLIP code and models, we use an open-sourced implementation, Stable unCLIP[1], in our experiments. Stable unCLIP provides two pretrained models conditioned on different CLIP image embeddings - **OpenAI CLIP ViT-L-14@224** [1] and **OpenCLIP ViT-H-14@224** [40], respectively. For evaluating the generality of our method within other CLIP backbones, we train another unCLIP model, conditioned on **SigLIP ViT-SO-14@384** [41], based on the above open-sourced implementation. Besides, we find in a preliminary toy experiment (in Section B.1) that the encoders of OpenAI CLIP ViT-L-14@224 have a similar embedding space to **OpenAI CLIP ViT-L-14@336** [1], therefore, for saving the computational cost of training new unCLIP models, we use the same existing Stable unCLIP for finetuning both of them. More details of these pretrained Stable unCLIP models are provided in Section B.2.

**un$^2$CLIP Training Details.** un$^2$CLIP is trained on 8 Nvidia-A100-40GB GPUs with a global batch size of 32, learning rate of 3e-7, using AdamW optimizer. For a fair comparison, we train un$^2$CLIP on the CC3M dataset [42] over 1 epoch following [21], taking around 15∼32 hours for different model types. The remaining hyper-parameters are kept the same as the training configuration of Stable unCLIP in the codebase.

**Compared Methods.** In addition to the **original** CLIP models, we mainly compare to the pioneering work **DIVA** [21] in this field that uses generative models to improve pretrained CLIP models. Different from ours, DIVA uses a pretrained text-to-image generative model as the backend, whose input space is misaligned to CLIP image encoders' output space. We also note that, most recently, there exists a *contemporaneous* work to ours named **GenHancer** [43]. GenHancer does not leverage existing well-trained generative models but trains imperfect generative models themselves for improving CLIP, which is different from DIVA and our un$^2$CLIP. More detailed discussions of the relationships and differences to DIVA and GenHancer are given in a separate section at Section A. In this section, for a more comprehensive comparison, we also present the results of GenHancer for reference but are dimmed in gray due to contemporaneity.

**Evaluated Tasks and Benchmarks.** We evaluate on several tasks to which CLIP has been applied and *require more detailed image understanding ability*, including the image-level MMVP-VLM benchmark [9] evaluating whether some detailed visual patterns are successfully captured (Section 4.2), the pixel-level dense vision-language inference task [2, 3, 4] (Section 4.3), and multimodal large language model related tasks [6, 7] (Section 4.4). Detailed introductions to these tasks and benchmarks are provided in corresponding subsections. For completeness, we also present the results on the *classical evaluation tasks* of CLIP, i.e., zero-shot classification and retrieval, in Section 4.5.

---

[1]Url: https://github.com/Stability-AI/stablediffusion/blob/main/doc/UNCLIP.MD. Due to built upon the text-to-image model Stable Diffusion [28], Stable unCLIP has an additional conditional text encoder. We feed it the empty string during un$^2$CLIP training.

## 4.2 CLIP-Blind Pair (MMVP-VLM Benchmark) Evaluation

We first evaluate our finetuned CLIP models on the MMVP-VLM benchmark [9]. The benchmark covers 9 visual patterns, each comprising 15 image pairs (30 images) accompanied by textual descriptions. The image pairs are collected in an adversarial manner to the original CLIP model, which are proximate in CLIP feature space but distant in the feature space of a visual self-supervised model (DINOv2 [44]). Only if both of the images are assigned to the accurate captions can a pair be deemed a correct case.

The results are presented in Table 1. Our method achieves the best average performance across different CLIP models. Notably, un$^2$CLIP significantly outperforms the original CLIP models and the previous DIVA method. This indicates that un$^2$CLIP is a general and effective method in improving CLIP to distinguish more detailed visual differences in images. Our performance also matches and slightly outperforms the contemporaneous work GenHancer.

Table 1: **MMVP-VLM benchmark evaluation.** The benchmark contains 9 visual patterns that original CLIP models often misinterpret: ◐: Orientation and Direction, **Q**: Presence of Specific Features, ⟳: State and Condition, ↕⁑: Quantity and Count, ♥: Positional and Relational Context, 🎨: Color and Appearance, ⚙⁸: Structural and Physical Characteristics, **A**: Texts, 📷: Viewpoint and Perspective. † denotes our reproduced results using official codes correspondingly.

| CLIP Model | Resol. | #Params | Method | ◐ | Q | ⟳ | ↕⁑ | ♥ | 🎨 | ⚙⁸ | A | 📷 | Avg |
|---|---|---|---|---|---|---|---|---|---|---|---|---|---|
| OpenAI ViT-L-14 | $224^2$ | 427.6M | Original | 13.3 | 13.3 | 20.0 | 20.0 | 13.3 | 53.3 | 20.0 | 6.7 | 13.3 | 19.3 |
| | | | DIVA | 13.3 | 20.0 | 40.0 | 6.7 | 20.0 | 53.3 | 46.7 | 20.0 | 13.3 | 25.9 |
| | | | GenHancer | 13.3 | 33.3 | 33.3 | 20.0 | 6.7 | 73.3 | 46.7 | 20.0 | 40.0 | 31.9 |
| | | | **un$^2$CLIP** | 0.0 | 33.3 | 46.7 | 26.7 | 13.3 | 80.0 | 40.0 | 20.0 | 33.3 | **32.6** |
| OpenAI ViT-L-14 | $336^2$ | 427.9M | Original | 0.0 | 20.0 | 40.0 | 20.0 | 6.7 | 20.0 | 33.3 | 6.7 | 33.3 | 20.0 |
| | | | DIVA | 26.7 | 20.0 | 33.3 | 13.3 | 13.3 | 46.7 | 26.7 | 6.7 | 40.0 | 25.2 |
| | | | GenHancer | 6.7 | 20.0 | 33.3 | 20.0 | 6.7 | 73.3 | 53.3 | 26.7 | 26.7 | 29.6 |
| | | | **un$^2$CLIP** | 6.7 | 33.3 | 46.7 | 13.3 | 13.3 | 80.0 | 40.0 | 20.0 | 20.0 | **30.4** |
| OpenCLIP ViT-H-14 | $224^2$ | 986.1M | Original | 6.7 | 13.3 | 53.3 | 26.7 | 6.7 | 73.3 | 40.0 | 13.3 | 26.7 | 28.9 |
| | | | DIVA† | 13.3 | 13.3 | 53.3 | 26.7 | 6.7 | 73.3 | 46.7 | 13.3 | 26.7 | 30.4 |
| | | | GenHancer† | 13.3 | 6.7 | 46.7 | 20.0 | 33.3 | 80.0 | 26.7 | 40.0 | 33.3 | 33.3 |
| | | | **un$^2$CLIP** | 26.7 | 13.3 | 53.3 | 20.0 | 33.3 | 86.7 | 46.7 | 13.3 | 33.3 | **36.3** |
| SigLIP ViT-SO-14 | $384^2$ | 878.0M | Original | 20.0 | 26.7 | 60.0 | 33.3 | 13.3 | 66.7 | 33.3 | 26.7 | 53.3 | 37.0 |
| | | | DIVA | 26.7 | 33.3 | 53.3 | 26.7 | 13.3 | 80.0 | 40.0 | 26.7 | 46.7 | 38.5 |
| | | | GenHancer | 26.7 | 20.0 | 66.7 | 33.3 | 13.3 | 86.7 | 40.0 | 26.7 | 46.7 | 40.0 |
| | | | **un$^2$CLIP** | 20.0 | 20.0 | 60.0 | 46.7 | 26.7 | 73.3 | 40.0 | 26.7 | 60.0 | **41.5** |

## 4.3 Dense Vision-Language Inference Evaluation

Next, we evaluate on the dense-prediction semantic segmentation task, which is a pixel-level task, thus evaluating more the detail-capturing ability of CLIP models. The segmentation task also acts as a helpful tool for qualitatively visualizing the behavior of the improved CLIP models.

**Evaluation Setup.** We follow training-free open-vocabulary semantic segmentation works [2, 3, 4] to evaluate our method. By keeping the CLIP model frozen, this setting provides a good way for diagnosing the model's pixel-level understanding capabilities. The work [2] first proposes to apply the pretrained CLIP model [1] to zero-shot semantic segmentation, by comparing the local patches of image embeddings to the candidate text embeddings. The following works, including MaskCLIP [2], SCLIP [3], and ClearCLIP [4], modify the inference-time network architecture to improve the performance. We employ these different methods, which have different initial performances, to evaluate the generality of our improved CLIP by substituting the CLIP model with our finetuned one.

**Datasets and Metric.** Following [2, 3, 4], we employ the mean Intersection over Union (mIoU) metric and evaluate on eight datasets widely used for open-vocabulary semantic segmentation. These datasets can be categorized into two groups: (1) Without background category: PASCAL VOC20 (VOC20) [45], PASCAL Context59 (Ctx59) [46], COCO-Stuff (Stuff) [47], Cityscapes (City) [48], and ADE20K (ADE) [49]; (2) With a background category: PASCAL VOC (VOC21) [45], PASCAL Context (Ctx60) [46], and COCO Object (Object) [47].

Table 2: **Open-vocabulary semantic segmentation quantitative comparison.** Results of DIVA and GenHancer are obtained using official checkpoints. The CLIP backbone is OpenAI ViT-L-14@336.

| Segmentation Method | CLIP-Improve. Method | Without background class | | | | | With a background class | | | Average |
|---|---|---|---|---|---|---|---|---|---|---|
| | | VOC20 | Ctx59 | Stuff | City | ADE | VOC21 | Ctx60 | Object | |
| CLIP | Original | 11.7 | 3.4 | 1.7 | 2.5 | 0.9 | 7.7 | 2.9 | 3.3 | 4.3 |
| | DIVA | 12.0 | 3.4 | 1.7 | 2.5 | 1.0 | 7.7 | 2.9 | 3.3 | 4.3 |
| | GenHancer | 8.4 | 2.9 | 1.3 | 2.7 | 0.7 | 4.6 | 2.5 | 1.7 | 3.1 |
| | **un²CLIP** | **17.3** | **5.1** | **2.6** | **3.8** | **1.3** | **9.3** | **4.3** | **4.3** | **6.0** |
| MaskCLIP | Original | 24.7 | 10.1 | 7.3 | 10.3 | 6.1 | 21.8 | 9.2 | 12.1 | 12.7 |
| | DIVA | 25.7 | 10.4 | 7.6 | 10.4 | 6.3 | 22.4 | 9.5 | 12.6 | 13.1 |
| | GenHancer | 13.5 | 6.4 | 3.4 | 9.2 | 3.7 | 12.3 | 5.9 | 4.9 | 7.4 |
| | **un²CLIP** | **30.0** | **12.9** | **8.9** | **13.1** | **7.5** | **25.2** | **11.6** | **13.5** | **15.3** |
| SCLIP | Original | 37.3 | 12.7 | 8.5 | 10.2 | 4.6 | 28.7 | 11.9 | 14.9 | 16.1 |
| | DIVA | 37.7 | 12.8 | 8.5 | 10.3 | 4.6 | 28.9 | 11.9 | 15.0 | 16.2 |
| | GenHancer | 21.0 | 7.7 | 3.6 | 6.8 | 2.2 | 15.1 | 7.0 | 5.3 | 8.6 |
| | **un²CLIP** | **53.8** | **19.5** | **12.0** | **16.1** | **6.9** | **38.6** | **17.9** | **19.3** | **23.0** |
| ClearCLIP | Original | 72.4 | 26.0 | 18.1 | 22.8 | 14.2 | 42.6 | 23.2 | 27.1 | 30.8 |
| | DIVA | 72.3 | 25.9 | 18.1 | 22.7 | 14.0 | 42.6 | 23.2 | 27.1 | 30.7 |
| | GenHancer | 52.1 | 22.9 | 11.8 | 17.1 | 10.3 | 24.2 | 20.0 | 10.2 | 21.1 |
| | **un²CLIP** | **76.5** | **30.5** | **20.6** | **26.4** | **16.0** | **47.6** | **27.3** | **29.6** | **34.3** |

**Quantitative Results.** Table 2 summarizes the performance of different CLIP improvement methods using different open-vocabulary segmentation methods. We observe that our method un²CLIP achieves the best results across different datasets and segmentation methods, significantly improving the performance of using the original CLIP model. Moreover, it is worth noting that switching to our finetuned model can further achieve an average 3.5 mIoU improvement for the state-of-the-art ClearCLIP method. We also note that the previous method DIVA achieves smaller improvements, and the contemporaneous work GenHancer achieves performance drops on this task. The following visualization results may provide some reasons.

**Qualitative Results.** In Figure 2, we present a qualitative comparison of the original CLIP model and its improvements, using the ClearCLIP segmentation method. It can be seen that: (1) Although the overall semantics of the image is correctly predicted using the original model, there are many local noises (false predictions) in the segmentation results. This is because the original CLIP model is trained towards a global image understanding objective, neglecting image details. (2) DIVA's segmentation maps are very close to the original CLIP model, indicating that DIVA has a relatively conservative finetuning stride, which cannot significantly improve the pixel-level task's performance. (3) GenHancer helps eliminate some noise compared to the original, but it makes some original-correctly predicted pixels wrong (e.g., the first and third columns), resulting in the overall weak quantitative results. (4) After our detail-oriented improvement, part of the false predictions are eliminated, and the results become smoother, indicating that un²CLIP is an effective upstream finetuning approach for making CLIP models better at performing dense prediction tasks.

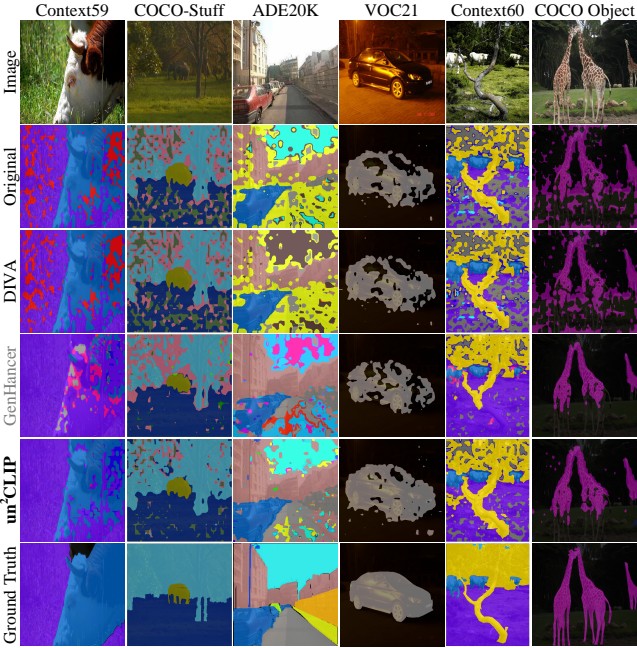

Figure 2: **Open-vocabulary semantic segmentation qualitative comparison.**

## 4.4 Multimodal Large Language Model Evaluation

Lastly, we evaluate whether our finetuned CLIP model can enhance the performance of MLLMs where CLIP serves as the vision encoder, with a particular focus on vision-centric benchmarks. To ensure fair comparisons, we adopt the evaluation setup in [21, 43] which test the improved CLIP using LLaVA-1.5 [7] without modifying LLaVA's default training configuration. We report results on the same set of benchmarks [9, 12, 50, 51, 52, 53] used in GenHancer [43] to simplify efforts in baseline reproduction. Due to the memory constrain of our computing resources, we conduct experiments based on the 7B-version Vicuna [54]. The results are presented in Table 3. We observe that the improved visual detail capturing ability of our finetuned CLIP also benefits MLLMs, leading to improved performance, particularly on vision-centric benchmarks.

Table 3: **MLLM benchmark evaluation.** Best and second best results are highlighted in **bold** and underline. Results on NaturalBench follow the official evaluation protocol [50], which differs from that in GenHancer [43], resulting in some missing entries. Baseline numbers are taken from [43].

| LLM | CLIP | | Vision-centric Benchmarks | | | | | | | | General Benchmarks | | | | |
|---|---|---|---|---|---|---|---|---|---|---|---|---|---|---|---|
| | | MMVP [9] | NaturalBench [50] | | | | CV-Bench 2D [12] | | CV-Bench 3D [12] | POPE [51] | | | SciQA-IMG[52] | Hallusion Avg. [53] |
| | | | Acc | Q-Acc | I-Acc | G-Acc | ADE20K | COCO | | rand | pop | adv | | |
| Vicuna-7B | Original | 24.7 | 67.3 | 37.7 | 43.8 | 12.7 | 49.6 | 60.9 | 58.7 | 87.3 | 86.1 | 84.2 | 66.8 | 27.6 |
| | DIVA | **31.3** | - | - | - | - | 51.3 | 63.4 | 60.2 | 87.9 | 87.0 | 84.6 | 66.3 | **28.6** |
| | GenHancer | 30.7 | - | - | - | - | 52.9 | 63.6 | **63.2** | **88.1** | 86.7 | 84.6 | 66.5 | 28.4 |
| | **un²CLIP** | **31.3** | **68.7** | **40.0** | **45.9** | **15.1** | **53.9** | **65.1** | 61.2 | 88.0 | **87.4** | **85.4** | **68.4** | 28.4 |

## 4.5 Zero-shot Classification and Retrieval Evaluation

For completeness, we evaluate the improved CLIP models on the two classical tasks: zero-shot image classification and zero-shot text-image retrieval. It is worth noting, however, that these tasks and their standard benchmarks (ImageNet-1K [55], CIFAR-10 [56], CIFAR-100 [56], Caltech-101 [57], SUN397 [58], FGVC Aircraft [59], Stanford Cars [60], Flickr30K [61], and COCO [62]), particularly those for image classification, are not designed to assess fine-grained visual understanding. Rather, they tend to favor models that capture high-level, category-discriminative features while ignoring subtle or classification-irrelevant details. This evaluation setup therefore contrasts with the main objective of our work, which is to enhance CLIP's ability to capture visual details as much as possible.

Table 4: **Zero-shot classification and retrieval evaluation.** Results of DIVA and GenHancer are obtained using official checkpoints. The CLIP model is OpenAI ViT-L-14@224.

| Method | Zero-shot Image Classification | | | | | | | Image-to-Text Retrieval@5 | | Text-to-Image Retrieval@5 | |
|---|---|---|---|---|---|---|---|---|---|---|---|
| | IN-1K | C-10 | C-100 | Cal-101 | SUN397 | Aircraft | Cars | Flickr30K | COCO | Flickr30K | COCO |
| Original | **75.5** | **95.6** | 75.9 | 86.7 | **67.6** | **31.7** | 77.9 | **97.3** | 79.4 | 87.3 | 61.0 |
| DIVA | **75.5** | 95.5 | **76.3** | **87.1** | 67.5 | 31.6 | **78.0** | **97.3** | **79.7** | 86.9 | 61.0 |
| GenHancer | 40.2 | 77.5 | 44.2 | 79.3 | 42.4 | 7.2 | 21.0 | 87.2 | 61.7 | 81.6 | 51.0 |
| **un²CLIP** | 62.4 | 89.0 | 65.6 | 86.8 | 59.2 | 22.0 | 63.3 | 96.4 | 77.6 | **90.1** | **65.5** |

As shown in Table 4, our finetuned model, which exhibits significantly enhanced performance on dense and vision-centric tasks in Section 4.2-4.4, exhibits a noticeable performance drop on zero-shot classification and comparable results on retrieval. Importantly, *this drop in classification accuracy does not necessarily reflect degraded image-text alignment*: Even on relatively simple datasets such as CIFAR-10, which contains only 10 coarse categories, the performance drop remains evident. In contrast, on segmentation benchmarks involving a much larger number of classes (e.g., 150-class ADE20K, 60-class Context), which also rely on the text encoder for label prediction, our fine-tuned model consistently outperforms the original CLIP (Table 2).

This discrepancy indicates that *the observed drop arises from task differences rather than image-text misalignment*. Classification tasks generally favor representations that emphasize dominant foreground semantics. However, somewhat "unfortunately" from the standpoint of classification, our finetuned model corrects many cases where the original CLIP mistakenly attends to background regions as salient foreground objects (see Figure 2). These corrections enhance visual detail understanding but lead to lower accuracy on conventional classification benchmarks.

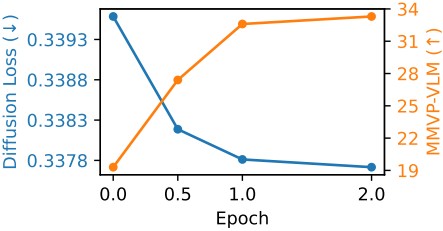

Figure 3: Diffusion loss and MMVP-VLM performance with respect to training epochs.

Table 5: **Ablation studies of un²CLIP.**

| Method | Diffusion Loss (↓) | MMVP-VLM (↑) |
|---|---|---|
| Original | 0.3396 | 19.3 |
| un²CLIP | 0.3378 | **32.6** |
| + Proj. layer (random init.) | 0.3441 | 16.3 |
| + Proj. layer (identity init.) | 0.3378 | 30.4 |
| + Proj. layer (two stage) | 0.3403 | 30.4 |
| + Updating generator $G$ | **0.3376** | 27.4 |

## 4.6 Ablation Studies

We conduct ablation studies of our method on the MMVP-VLM benchmark, using OpenAI ViT-L-14@224 as the baseline model. Before analyzing our method, we first introduce a diagnostic tool that measures the reconstruction ability of the CLIP-encoding generator-decoding pipeline.

**Diffusion Loss as a Diagnostic Tool.** We adopt the diffusion loss (Eq. (2), (5)) as a diagnostic metric to evaluate the reconstruction ability of different CLIP encoders $E$ (e.g., before and after finetuning). A lower loss indicates that the encoder captures more details from the input image, leading to a better reconstruction. To compute the expectation term $\mathbb{E}_{\mathbf{x},\epsilon,t}$ in practice, we random sample noises $\epsilon$ and timesteps $t$ (with $\mathbf{x}$ from the test set). However, since random sampling can affect result comparability, to make the results comparable, we pre-sample and store two sets of (20) random $(\epsilon, t)$ pairs for each $\mathbf{x}$, and use the same sets across all evaluation trials (e.g., before and after finetuning). This pre-storing strategy makes the metric analogous to the "test loss" used in [63, 64]: although it shares the same formulation as the training objective (Eq. (5)), it is computed with fixed randomness and test images. In contrast, the training loss is unsuitable for comparison because $\epsilon$ and $t$ vary across iterations, leading to non-comparable results.

**Does Better Reconstruction Lead to Better Recognition?** To answer this question, we first train our default un²CLIP over different epochs and plot their losses and MMVP-VLM performances. As presented in Figure 3, we observe that models with smaller losses achieve better recognition performances. The reason behind this is that the diffusion loss is a lower bound of the generative models' likelihood $\mathbb{E}_{\mathbf{x}}\left[\log p_G(\mathbf{x}|E(\mathbf{x}))\right]$, which directly relates to our finetuning goal of capturing more visual details, as introduced in Section 3.2. Notably, since calculating the diffusion loss does not require task labels, it can be used as a tool for predicting the tendency of task performance when using our *default* design[2].

**Introducing Projection Layers.** In the previous work DIVA [21], a linear projection layer is inserted between the CLIP image encoder and the generative model, because DIVA uses a pretrained text-to-image model whose input space mismatches the CLIP image embedding in terms of embedding dimension and semantics. The projection layer is trained together with CLIP during finetuning. We investigate the impact of introducing a projection layer into our framework. Since the input space of unCLIP has already been aligned with CLIP image embeddings, inserting a projection layer is actually not needed in our framework and may break the alignment. Our experiments verified this. As shown in Table 5, we first conduct an experiment using *randomly initialized* weights for the linear projector. Since the inserted projector alters the data flow in the pretrained CLIP-unCLIP model, it achieves a higher loss than the original, causing downgraded performance. We further modify the initialization to be an *identity* weight matrix with *zero* bias, making it as if the projector does not exist at the beginning of finetuning. This modification enables the framework to make progress, as shown in Table 5. However, once the projector is updated, the alignment between the encoder's output and the generator's input does not hold, and it may taking away part of the knowledge learned from the generator, leading to suboptimal performance compared to our default method. We also experimented with a *two-stage training strategy* for the projector, following [6], where the projector is trained first and then frozen during the image encoder finetuning. This approach yields slightly worse performance than our default setting. We attribute this to the fact that projection layers are unnecessary in our framework, as the image encoder and generator are already aligned in unCLIP. Even when the projector is pretrained separately, it remains challenging for it to fully reproduce the naturally aligned encoder-generator environment established in unCLIP.

---

[2]The conclusion would not hold if not following our default design, as shown in the following ablations.

**Updating the Generator $G$.** As mentioned in Section 3.2, we set the unCLIP generator to be frozen during the finetuning process, to encourage the finetuned encoder's outputs stay close to the original unCLIP's input space, i.e., the original CLIP image-text embedding space. Here, we examine the impact of updating the generator together with the encoder. The result is presented in the last row of Table 5. It can be seen that the full finetuning achieves the best reconstruction, as there are more parameters that can be tuned. However, similar to the issue of the above-introduced learnable projector, updating the generator causes the finetuned encoder to move away from the original embedding space, resulting in a performance drop. In such cases, we cannot expect better recognition performances when observing better reconstructions.

**Visualization Analysis.** We give a visualization analysis of our default method and some representative ablations. Specifically, we use the finetuned encoder to perform the encoding-decoding pipeline on an input image, thereby visualizing which features of the image are successfully captured. When sampling from different models, we use the same initial random noise to make the visualization results comparable. The results are shown in Figure 4. First, by comparing generated images of the original CLIP and un²CLIP, it can be seen that after un²CLIP finetuning, the main patterns of the images are successfully captured, such as orientation of the first example and spatial position of the second example. Longer finetuning achieves better qualitative reconstructions, which aligns with the quantitative result in Figure 3. On the

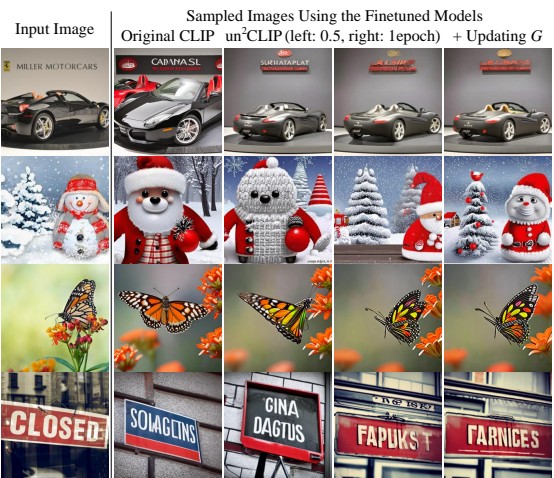

Figure 4: unCLIP generated images using original and finetuned CLIP models.

other hand, updating $G$ achieves a visually comparable or slightly better reconstruction than our default (e.g., row 2, a better shape of the snowman), but in this case a better reconstruction does not means a better recognition performance, as analyzed in the previous paragraph.

## 5   Conclusion

In this paper, we propose an image-based CLIP finetuning method un²CLIP to address the problem that pretrained CLIP models fall short in capturing visual details. By inverting a generative model that takes the CLIP image embedding as input, our method enables the finetuned CLIP to gain knowledge from the powerful generative model while preserving the alignment to its original embedding space simultaneously. Our method is simple yet effective, based on the key finding that the existing unCLIP generative model fits exactly to our goal. Extensive experiments across image and pixel level tasks demonstrate that by changing the original CLIP to our finetuned one, the performance of tasks to which CLIP has been applied and require visual detail capturing can be significantly improved, such as open-vocabulary semantic segmentation and vision-centric multimodal understanding.

**Limitations.** A potential limitation of this work is that finetuning CLIP requires a pretrained unCLIP model at first. Thankfully, the community has provided some pretrained unCLIP models, which are built upon widely used CLIP backbones. But if we take the computational cost of training the unCLIP into consideration, such an additional cost may be tolerable, given that improving the foundation model CLIP is an upstream work that could benefit many vision-centric downstream tasks where CLIP has been applied.

## Acknowledgments

This work is partially supported by National Science and Technology Major Project 2021ZD0111901 and National Natural Science Foundation of China 62376259.

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

# A    Relationships and Differences to Prior and Contemporaneous Works

This paper leverages generative models to improve CLIP's visual detail capturing ability. There are two related works in this direction – one prior [21] and one contemporaneous [43]. We discuss the relationships and differences with them below. An overview is summarized in Table 6.

Table 6: **Comparison with related works.** $E$ stands for the CLIP image encoder and $G$ stands for the generative model. *: with the generative model frozen.

| Work | Architecture | | | Findings |
| --- | --- | --- | --- | --- |
| | Generative Model | Projector | Aligned with CLIP? | |
| DIVA [21] | Large-scale pretrained text-conditioned model | Trainable with $E$ | ✗ | - |
| GenHancer [43] | Small-scale projector-incorporated model trained from scratch | Pretrained with $G$ | ✓ | Reconstruction ↑ ⇏ Recognition ↑ |
| un$^2$CLIP | Large-scale pretrained CLIP-image-embedding-conditioned model | Free | ✓ | Reconstruction ↑ ⇒ Recognition ↑ * |

**Comparison with Prior Work DIVA [21].** DIVA is a pioneering method in this line of research. It employs a pretrained text-to-image generative model, Stable Diffusion [28], as the backend to improve CLIP. Since the output space of the CLIP image encoder is not aligned with the conditional input space of Stable Diffusion, both in terms of embedding dimensionality and semantics, DIVA inserts a projector between the CLIP encoder and the generative model. The projector is trainable during CLIP finetuning, which may take away part of the knowledge learned from the generator. Different from DIVA, we use the unCLIP generative model, which takes the CLIP image embedding as input, thereby achieving a projector-free framework, enabling a seamless and effective process for CLIP encoder enhancement.

**Comparison with Contemporaneous Work GenHancer [43].** GenHancer is a contemporaneous work that appeared online on March 25th, 2025. Different from DIVA and our un$^2$CLIP that use existing well-trained generative models, GenHancer trains generative models from scratch specifically to improve the CLIP, and draws a main conclusion that "perfect generation (reconstruction) does not always yield desirable visual representations". We analyze the key relationships and differences below:

**(1) Relationship in Architecture: Projector-incorporated vs. Projector-free.** Similar to DIVA but different from ours, GenHancer introduces a projector between the CLIP encoder and the generative model. Unlike DIVA, the projector in GenHancer is pretrained together with the generative model in a pretraining stage, prior to CLIP finetuning. This design can be interpreted as the projector is incorporated into the generative model, allowing the generative model's input to align with CLIP's output at the CLIP finetuning stage. Viewed this way, both GenHancer and our un$^2$CLIP achieve an aligned, seamless encoder-generator pipeline, which facilitates more effective enhancement of the CLIP image encoder.

**(2) Difference in Architecture: Capability of the Generative Model.** GenHancer does not utilize existing large-scale pretrained generative models. Instead, it trains lightweight generative models from scratch, using the CC3M dataset [42] for a single epoch. In contrast, our method leverages the pretrained Stable unCLIP model, which is built upon Stable Diffusion and trained with approximately 200,000 A100 hours[3]. This pretrained backbone offers substantially greater model capacity. Given that our optimization objective (Eq. (4)) is accurately estimating the conditional distribution $p(\mathbf{x}|E(\mathbf{x}))$, a more capable generator provides a stronger training signal for improving the CLIP encoder. Moreover, using a well-trained generator enables visualization of how much the encoder has improved, and could provide intuitive explanations for the prediction behaviors of the models, through the generated images via the encoding-decoding pipeline, as demonstrated in Figure 4 of the main paper and Figure 5 of this appendix, compared with Figure 4 in GenHancer.

**(3) Divergent Findings: Relationship between Reconstruction and Recognition Performance.** GenHancer reports that "perfect generation (reconstruction) does not always yield desirable visual

---

[3]https://huggingface.co/stabilityai/stable-diffusion-2-1-unclip#environmental-impact

representations" in its Figure 1. Our conclusion differs, as shown in the ablation studies in Section 4.6 of the main paper. The key distinction lies in how reconstruction quality is measured. We use the diffusion loss, which is the lower bound of the generative model's likelihood, as the measurement for reconstruction as well as our finetuning objective. A lower diffusion loss indicates that more information is preserved in the encoding-decoding process. Therefore, as for our framework with freezing the generator during the encoder's finetuning, better reconstruction leads to improved recognition performance in visual detail tasks, as analyzed in Section 4.6. On the other hand, GenHancer uses CLIP score [65] as the measurement for both generation and reconstruction. Since the CLIP score primarily measures image-text alignment rather than pixel-level fidelity, it is more effective for evaluating text-faithfulness of generated images of text-to-image models, but suboptimal for evaluating visual reconstruction quality of image-to-image encoding-decoding pipelines.

## B  Additional Experimental Details

### B.1  A Toy Experiment Validating Alignments between OpenAI ViT-L-14@224 and ViT-L-14@336

As described in Section 4.1 of the main paper, we find that the image and text encoders of OpenAI CLIP ViT-L-14@224 and OpenAI CLIP ViT-L-14@336 have a similar embedding space. Therefore, we use the pretrained Stable unCLIP for OpenAI CLIP ViT-L-14@224 to improve both of them, thereby avoiding the cost of training a separate unCLIP model for OpenAI CLIP ViT-L-14@336.

This observation is based on the following toy experiment, in which we swap the image and text encoders between the two CLIP models and evaluate the resulting combinations. We use the zero-shot classification task for this experiment. The results are summarized in Table 7. We find that (1) the two original models have similar performance, and (2) after swapping their image and text encoders, the hybrid models retain comparable performance to the originals. These results suggest that the embedding spaces of the two models are closely aligned. Therefore, for efficiency, we use the pretrained unCLIP model for OpenAI CLIP ViT-L-14@224 to improve both of them during our un$^2$CLIP training.

Table 7: **Zero-shot classification performance of image-text encoder swapped CLIP models.** The two CLIP models are OpenAI CLIP ViT-L-14@224 and OpenAI CLIP ViT-L-14@336.

| Image Encoder | Text Encoder | IN-1K [55] | C-10 [56] | C-100 [56] | Cal-101 [57] | SUN397 [58] | Aircraft [59] | Cars [60] |
|---|---|---|---|---|---|---|---|---|
| @224 | @224 | 75.5 | 95.6 | 75.9 | 86.7 | 67.6 | 31.7 | 77.9 |
| @336 | @336 | 76.6 | 94.9 | 74.4 | 87.2 | 68.7 | 33.4 | 79.3 |
| @224 | @336 | 75.4 | 95.5 | 76.0 | 86.7 | 67.7 | 31.5 | 78.0 |
| @336 | @224 | 76.5 | 95.0 | 74.5 | 87.2 | 68.5 | 33.4 | 79.4 |

### B.2  Pretrained Stable unCLIP models

We use the code (MIT License)[4] and models (CreativeML Open RAIL++-M License)[5] of Stable unCLIP to implement our method. This release includes two pretrained unCLIP models, conditioned on OpenAI CLIP ViT-L-14@224 [1] and OpenCLIP ViT-H-14@224 [40] image embeddings. These two models are finetuned versions of the `stable-diffusion-2-1` model[6], adapted to accept CLIP image embeddings as conditional inputs. Stable Diffusion is a type of latent diffusion model [28] that performs denoising in the latent space of a pretrained KL-VAE [66] model. The latent size of `stable-diffusion-2-1`, and therefore of the two unCLIP models, is 96×96×4. The details of these models are summarized in Table 8. As explained in Section 4.1 and Section B.1, we use the same unCLIP model to improve both the OpenAI CLIP ViT-L-14@224 and OpenAI CLIP ViT-L-14@336 image encoders during our un$^2$CLIP training stage.

As introduced in Section 4.1, to evaluate the generality of our approach across different CLIP backbones, we additionally train a new Stable unCLIP model conditioned on SigLIP ViT-SO-14@384 [41]

---

[4]https://github.com/Stability-AI/stablediffusion/blob/main/doc/UNCLIP.MD
[5]https://huggingface.co/stabilityai/stable-diffusion-2-1-unclip
[6]https://huggingface.co/stabilityai/stable-diffusion-2-1

Table 8: **Model hyper-parameters of Stable unCLIP and training costs.**

| Conditional Branch | | Denoising Backbone | | unCLIP | un$^2$CLIP |
|---|---|---|---|---|---|
| CLIP Image Encoder | #Params | Input Size | #Params | Training Cost | Training Cost |
| OpenAI CLIP ViT-L-14@224 | 303 M | 96×96×4 | 869 M | Pretrained | 30h |
| OpenAI CLIP ViT-L-14@336 | 304 M | 96×96×4 | 869 M | N/A | 30h |
| OpenCLIP ViT-H-14@224 | 632 M | 96×96×4 | 870 M | Pretrained | 32h |
| SigLIP ViT-SO-14@384 | 428 M | 64×64×4 | 869 M | ~5d | 15h |

image embeddings, based on the open-source implementation mentioned above. To reduce training cost, this model is built upon `stable-diffusion-2-1-base`[7], which operates in a smaller latent space of 64×64×4. We train this unCLIP model on the CC3M dataset [42], using a global batch size of 2048 following the configuration of `stable-diffusion-2-1-base`. The model is trained for 15K iterations (about 10 epochs over CC3M), taking about 5 days with 8 Nvidia-A100-40GB GPUs, as summarized in the last row of Table 8.

### B.3 Computational Costs

The training cost of un$^2$CLIP in each experiment is reported in the rightmost column of Table 8, where the SigLIP experiment, due to the smaller input size to the main denoising network, has a relatively faster training speed. The full research project, including some preliminary, failed, ablative, and downstream task experiments, takes about 3 ~ 4 times the sum of the reported training costs.

## C Additional Ablation Studies

### C.1 un$^2$CLIP Training (CLIP Finetuning) Dataset

By default, we follow DIVA [21] and use CC3M [42] as the training dataset for fair comparison. We also note that the concurrent work GenHancer [43] adopts the same setting. However, none of these works investigate the effect of the training dataset choice, leaving this as an open question. In this section, we study how the choice of training dataset within our un$^2$CLIP framework affects the generalizability of the finetuned model. For this ablation, we evaluate not only on the MMVP-VLM benchmark but also on open-vocabulary segmentation tasks to comprehensively assess generalization.

**Training on ImageNet-1K.** We first replace CC3M with ImageNet-1K [55], a class-balanced dataset containing ~1.3M images across 1000 categories. We keep the total number of training iterations the same as with CC3M to isolate the effect of dataset content. As shown in Table 9 (segmentation results averaged over 8 datasets), the ImageNet-finetuned model achieves competitive results with the CC3M-finetuned one on MMVP-VLM. This may be because the MMVP-VLM dataset is relatively small, and its image patterns can be well covered by both CC3M and ImageNet training sets. In fact, part of MMVP-VLM's images originate from ImageNet, as introduced in [9].

However, on segmentation benchmarks, the ImageNet-finetuned CLIP does not yield consistent improvements. Notably, ImageNet-1K is a highly structured dataset (e.g., class-balanced, lacking the "person" class), which differs from web-collected datasets such as CC3M, CLIP's pretraining dataset WebImageText [1], and Stable unCLIP's training dataset LAION-5B [67]. *This raises the question of whether it is the dataset content (distribution) or dataset scale that results in poor generalizability.*

**Training on 10% CC3M images.** To further investigate this question, we train another model using a randomly sampled 10% subset of CC3M (~0.3M images), even smaller than ImageNet-1K in scale. As shown in the last row of Table 9, this model exhibits better generalization than the ImageNet-finetuned one, suggesting that *dataset distribution* (e.g., source diversity and similarity to CLIP's pretraining data) *plays a more crucial role than dataset scale* in achieving better generalized finetuned CLIP models.

---

[7]https://huggingface.co/stabilityai/stable-diffusion-2-1-base

Table 9: Ablation study on **un$^2$CLIP** training datasets.

| Method | un$^2$CLIP Training Dataset Scale | MMVP-VLM | CLIP Seg. | MaskCLIP Seg. | SCLIP Seg. | ClearCLIP Seg. |
|---|---|---|---|---|---|---|
| Original CLIP | N/A | 19.3 | 5.3 | 14.7 | 24.0 | 34.4 |
| un$^2$CLIP - CC3M (default) | ∼3 M | **32.6** | **5.8** | **15.6** | **25.9** | **34.8** |
| un$^2$CLIP - ImageNet-1K | ∼1.3 M | **32.6** | 5.3 | 13.9 | 24.0 | 33.0 |
| un$^2$CLIP - 10% CC3M | ∼0.3 M | 31.1 | **5.8** | 15.3 | **25.9** | 34.5 |

## C.2 Training with Image-Text Data

In addition to the generator-frozen strategy, we further explore using image-text supervision to mitigate the potential language-shift problem during image encoder finetuning. Specifically, we incorporate an additional image-text loss into our training objective, updating both the image encoder $E$ and the generator $G$. This loss is implemented as the negative cosine similarity between image and text features from the CLIP encoders, using paired image-text data from CC3M. Jointly optimizing the two objectives introduces a balancing hyperparameter between the default unCLIP-inversion loss and the introduced image-text loss, for which we test two values (1 and 0.1).

Before conducting experiments, *we note that the added image-text objective may reintroduce the very issue this paper aims to address*: captions often describe coarser-grained semantics than images themselves, potentially making the finetuned model less sensitive to visual details. Results are summarized in Table 10. We observe that: (1) Simply combining the two losses with equal weight (#4) causes a performance drop compared to not introducing the image-text loss (#3), suggesting that the added supervision indeed hampers the model's ability to capture visual details, as hypothesized; and (2) Using a smaller weight for the image-text loss (#5) yields better results, indicating a more balanced trade-off between visual-detail capturing and image-text alignment during optimization. However, #5 still falls short of #2. These results demonstrate that our default strategy, leveraging the image-text aligned space of unCLIP and updating only the encoder, remains *simple* (no additional losses or hyperparameter tuning), *effective* (achieving the best performance), and *efficient* (fewer trainable parameters).

Table 10: Effect of **incorporating image-text supervision** during un$^2$CLIP training.

| # | Method | un$^2$CLIP Training Loss | MMVP-VLM |
|---|---|---|---|
| 1 | Original CLIP | N/A | 19.3 |
| 2 | Default (Training $E$ only) | unCLIP-inversion loss | **32.6** |
| 3 | Training $E$ and $G$ | unCLIP-inversion loss | 27.4 |
| 4 | Training $E$ and $G$ | unCLIP-inversion loss $+ 1 \times$ image-text loss | 25.9 |
| 5 | Training $E$ and $G$ | unCLIP-inversion loss $+ 0.1 \times$ image-text loss | 28.9 |

# D  Additional Qualitative Results

## D.1 MMVP-VLM Visualization Results

Figure 5 presents qualitative examples of the MMVP-VLM benchmark. For each case, we also apply the CLIP-unCLIP encoding-decoding pipeline to both the original and our improved CLIP models, as done in the visualization analysis paragraph in Section 4.6 of the main paper. These generated images help provide intuitive explanations for the prediction behaviors of the models – illustrating why the original (our improved) CLIP model makes incorrect (correct) predictions for some cases.

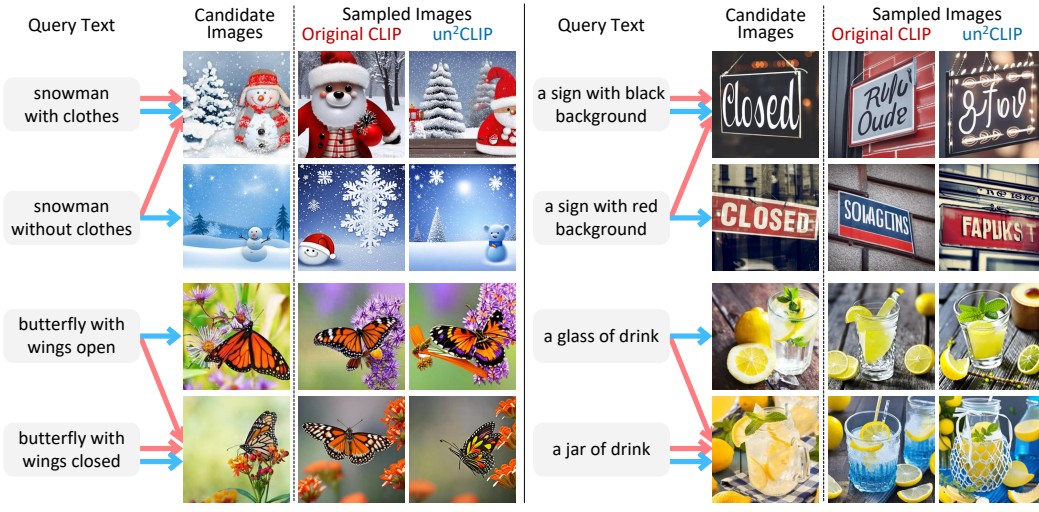

Figure 5: **MMVP-VLM visualization results.** Predictions of the original CLIP and our improved CLIP are shown with red and blue arrows, respectively. Generated images using the CLIP-unCLIP encoding-decoding pipeline are shown at right, providing visual insight into each model's predictions.

## D.2 MLLM Visualization Results

Figure 6 presents qualitative examples of MLLM tasks, focusing on vision-centric benchmarks.

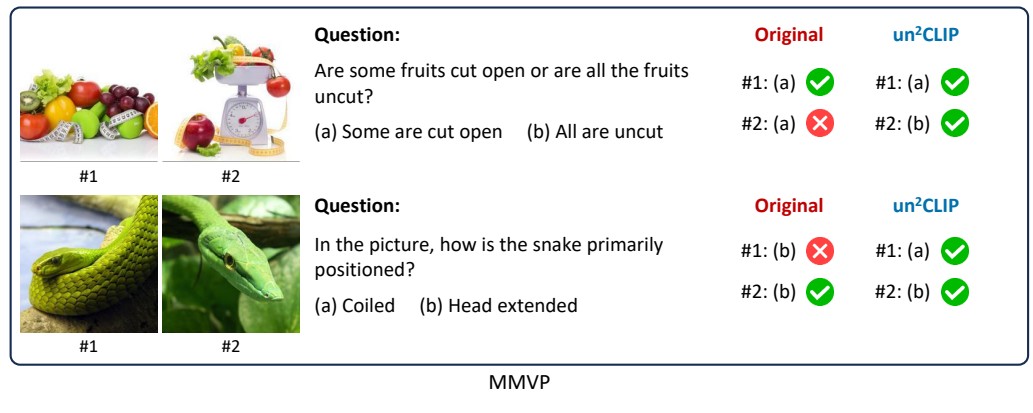

MMVP

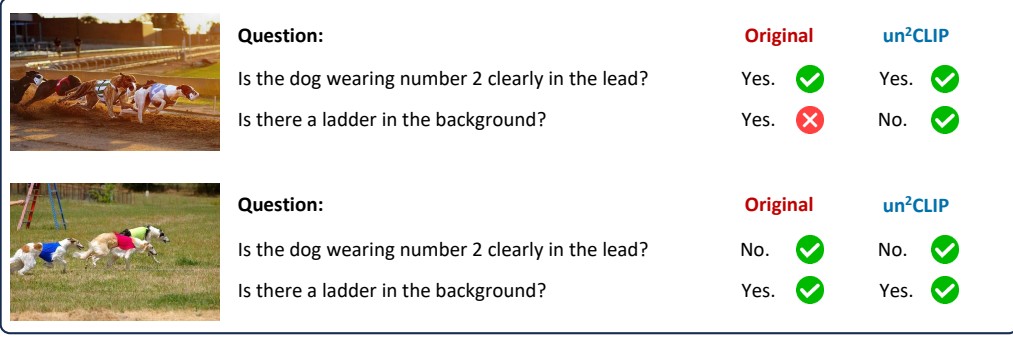

NaturalBench

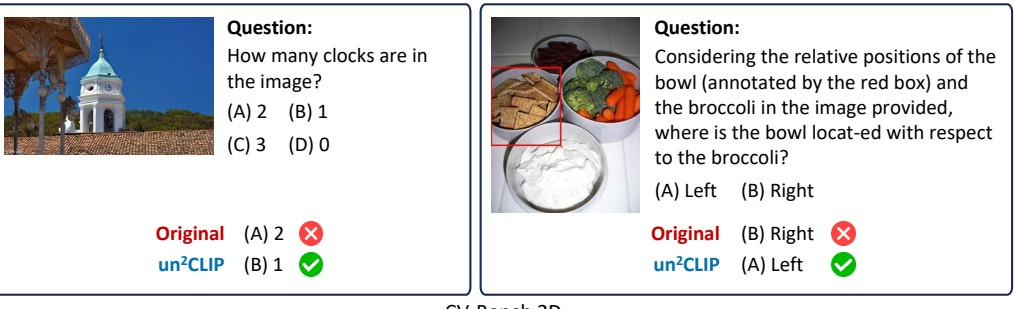

CV-Bench 2D

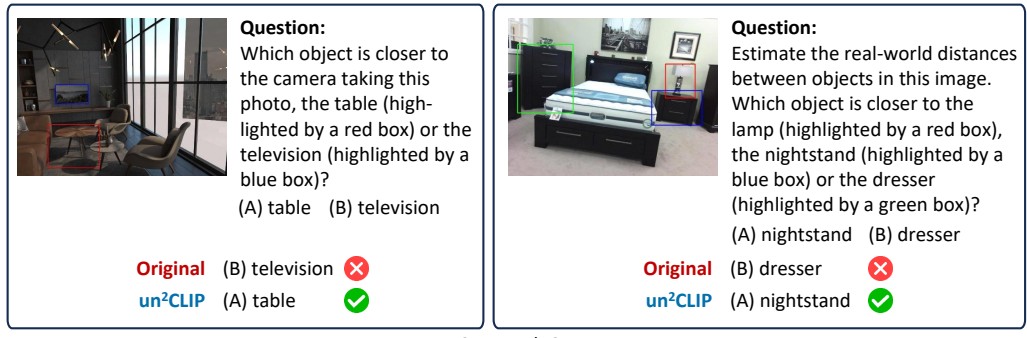

CV-Bench 3D

Figure 6: **Vision-centric MLLM visualization results.**

