# OpenReview forum: "un$^2$CLIP: Improving CLIP's Visual Detail Capturing Ability via Inverting unCLIP"
_NeurIPS.cc/2025/Conference — NeurIPS 2025 poster_

### Official Review · Reviewer_dkG9 · 2025-06-05

**Clarity:** 3
**Significance:** 3
**Originality:** 2
**Rating:** 4
**Confidence:** 3

**Summary:**

This paper proposes to improve the fine-grained detail representation capability of a CLIP with a reconstruction loss. The proposed method, un^2CLIP, achieves this by introducing the image generator of unCLIP (a previous work that learns a reconstruction model upon the clip feature). The original image encoder is fine-tuned with image data only, using the reconstruction loss as guidance.  The authors conducted evaluation on MMVP-VLM, open-vocabulary segmentation and MLLMs. The evaluations and ablation studies justified the design of the proposed method.

**Questions:**

* The authors freeze the image generator and only fine-tune the image encoder, so as to reduce language-shift when training the model with only image data. How about jointly tuning the encoder and generator with text-image data (since the training was conducted on CC-
3M with image-text pairs)? I see the ablation in Table 4, but it seems the ablation was still conducted with image data only?

* This work claims the freeze-generator-finetune-encoder training method reduces the language-shift problem, making the embedding space of the finetuned model still lie in the original space. How is the performance of the the fine-tuned model on zero-shot recognition/text-image retrieval? More evaluation on this would be helpful to justify this claim.

* Some properties of the un^2CLIP are not evaluated. For example, in the concurrent work GenHancer, the performance of generation and understanding with respect to training iterations are different: better reconstruction does not always lead to better understanding. Does this also apply to the proposed method? More experiments are expected to better showcase the property of the proposed method.



**Minor:**

It is not very clear to me how the diffusion loss is used as the indicator of the training difficulty. It would be better if the authors could elaborate on this at somewhere in the paper or the appendix.

**Ethical Concerns:**

["NO or VERY MINOR ethics concerns only"]

**Final Justification:**

Most of my concerns are addressed during the rebuttal period. I thus keep my original rating that the paper is slightly above the accepetance threshold.

**Limitations:**

yes

**Quality:**

3

**Strengths And Weaknesses:**

**Strengths:**

* The problem studied in this work is interesting. The poor fine-grained visual representation capability of CLIP's visual encoder has long been a problem in the MLLM community.

* The proposed method is straightforward, easy to understand and implement.


**Weaknesses:**

* The proposed method relies on a pretrained image generator from unCLIP, which limits its generalization to other CLIP encoders since not every CLIP model has a corresponding image generator.

* Some evaluations are missing, such as training with image-text pair, evaluation on CLIP related benchmarks (see questions below).

* Utilizing generation models to improve the representation capability of an image encoder is not a very new paradigm, the novelty of the proposed method is relatively limited.


Overall, I think this paper is good and interesting, but the novelty seems to be relatively limited. Adding more experiments can be helpful to justify the claims in the paper and help the readers better understand the property of the proposed method.

---

> ### Author Rebuttal · Authors · 2025-07-31
>
> We thank the reviewer for the detailed review and helpful feedback. We are encouraged that the reviewer finds the paper overall good and interesting, and the method easy to understand and implement.
>
> Below, we provide detailed point-by-point responses. We are more than willing to provide additional clarification should any questions remain.
>
> > **W1. The proposed method relies on a pretrained image generator from unCLIP, which limits its generalization to other CLIP encoders since not every CLIP model has a corresponding image generator.**
>
> Relying on a pretrained unCLIP image generator is indeed a limitation of our approach and we have summarized it in the last paragraph of Section 5. There are two solutions to address this issue, as detailed in Appendix B and mentioned in Section 4.1:
>
> (1) A trick partial solution using shared generators. We find that some existing CLIP models (e.g., OpenAI CLIP ViT-L-14@224 and ViT-L-14@336) have highly similar embedding spaces thus they can share a same unCLIP image generator.
>
> (2) A general solution to train corresponding unCLIP image generators. Training an unCLIP generator for a specific CLIP encoder is feasible and not overly expensive because there are some pretrained powerful generative models in the community such as Stable Diffusion that unCLIP can built upon. In the codebase used in our experiments, the unCLIP models are finetuned versions of Stable Diffusion. And the unCLIP model for the SigLIP used in the experiment is trained by ourselves, taking about 5 days on 8 A100-40G gpus. Such costs may be tolerable, given that improving the foundation model CLIP is an upstream work that could benefit many vision-centric downstream tasks where CLIP is applied.
>
> ---
> > **W3. Utilizing generation models to improve the representation capability of an image encoder is not a very new paradigm.**
>
> We agree with the reviewer's comment that using generative models to improve CLIP image encoders is not an entirely new paradigm, as we share a high-level similarity with the prior work DIVA in this point. Our key contribution lies in *identifying and addressing the encoder-generator mismatch issue* of the previous work, through *finding* that an existing type of generative models (*unCLIP*) *offers a more seamless and effective* CLIP enhancement framework, and demonstrating better performance on various benchmarks.
>
> ---
> > **Q1 (W2). Jointly tuning the encoder and generator with text-image data.**
>
> Thank you for this suggestion which helps broaden the scope of our ablation study.
>
> **Implementation.** Following the reviewer's advice, we incorporate an additional image-text loss into our training objective, updating both the image encoder $E$ and the generator $G$. This loss is implemented as the negative cosine similarity between the image and text features from the CLIP encoders, using paired training data from CC3M. Jointly optimizing two losses introduces a hyperparameter $\lambda$ that balances the previous unCLIP-inversion loss and the image-text loss. We tried two values for $\lambda$ (1 and 0.1) during this tight rebuttal period.
>
> **Pre-analysis.** *We note that the added image-text objective may re-introduce the issue that this paper aims to address*: captions often reflect coarser-grained semantics than images themselves, making the finetuned model capture less visual details.
>
> **Results.** Results are reported in the last two rows of the table below. We observe that:
> - Simply adding the two losses with equal weight (#4) causes a performance drop compared to not introducing the image-text loss (#3), suggesting that the introduced image-text loss negatively impacts the model’s ability to capture visual details, as hypothesized.
> - Using a smaller weight for the image-text loss (#5) yields a better result, indicating a more balanced trade-off between visual-detail capturing and image-text-alignment during optimization.
>
> However, #5 still falls short of #2. These results demonstrate that our default strategy, leveraging the image-text aligned space of unCLIP and updating only the encoder, remains *simple* (no other loss terms or tuning trade-offs), *effective* (current the best performance), and *efficiency* (fewer trainable parameters).
>
> |#|Method|un$^2$CLIP Training Loss|MMVP-VLM|
> |-|:-|-|:-:|
> |1|Original CLIP|N/A|19.3|
> |2|un$^2$CLIP default (training $E$ only)|unCLIP-inversion loss|**32.6**|
> |3|Training $E$ and $G$|unCLIP-inversion loss|27.4|
> |4|Training $E$ and $G$|unCLIP-inversion loss + 1 × image-text loss|25.9|
> |5|Training $E$ and $G$|unCLIP-inversion loss + 0.1 × image-text loss|28.9|
>
> ---
> > **Q2 (W2). How is the performance on zero-shot recognition/text-image retrieval? Evaluation on this would be helpful to justify the image-text-alignment claim.**
>
> The zero-shot classification and text-image retrieval results are in Table 8 in Appendix C, where we observe notable performance drop on most classification datasets, and maintained text-image retrieval performance compared to the original CLIP model.
>
> However, we argue that **the drop in classification accuracy does not necessarily indicate a degradation in image-text alignment**. Because even on relatively *"easy" classification dataset* like CIFAR-10 that contains only 10 coarse classes, the performance *drop is evident*. In contrast, on *various segmentation benchmarks that contain more classes* (e.g. 150-class ADE, 60-class Context) which also rely on the text encoder for prediction, our finetuned model *consistently outperforms* the original CLIP.
>
> This discrepancy suggests that **the classification drop stems from task differences, not broken alignment**. As discussed in our responses to Reviewer j8dh's last comment and Reviewer dYip's comment 2, classification tasks tend to favor models that focus on foreground semantics. And "unfortunately" our finetuned model corrects many mispredictions where CLIP mistakenly classifies background regions as salient foreground objects (see Fig. 2). These corrections improve visual detail understanding but degrade performance on the conventional classification task.
>
> Importantly, if the image-text alignment is truly broken, the finetuned model would also fail on the segmentation tasks, which it does not. This supports the claim that the alignment remains intact. We will move the classification/retrieval experiments and analysis from the appendix to the maintext in the revised paper to highlight this point.
>
> ---
> > **Q3. Does better reconstruction lead to better understanding?**
>
> We also noticed this question and it is studied in Paragraph 3 of Section 4.5 in our paper, where we draw a **different conclusion** - better reconstruction in our *default*\* framework always lead to better understanding (recognition) - to the concurrent work GenHancer.
>
> The key distinction causing the different conclusion lies in *how reconstruction is measured*. In GenHaner paper's Fig. 1, "reconstruction" is treated as "generation", and evaluated using the CLIP score which is often used to evaluate text faithfulness of generated images of text-to-image models. However, this metric is arguably unsuitable for evaluating "reconstruction" in the image-only encoding-decoding frameworks. In contrast, we use a more task-relevant metric - the estimation of Eq. (4) to measure reconstruction (the introduced diffusion loss in Section 4.5, further clarified in our response to the next question). Since Eq. (4) directly connects to the goal of our problem, it is reasonable that better reconstruction (lower loss) leads to better recognition. These are summarized in the last paragraph of Appendix A.
>
> P.S. For a more figurative verification of our conclusion, we can see the visualization result in Fig. 4 (Page 9) of our paper, where the middle columns of images (using un$^2$CLIP finetuned CLIP encoder) achieve better reconstructions of the input images, compared to the original CLIP encoder. And un$^2$CLIP also achieves better quantitative results shown in Table 1 or 4.
>
> \* "default" here refers to not using other training configurations such as also updating the generator introduced in our ablation studies. This point is also clarified in Footnote 2 of our paper.
>
> ---
> > **Minor. How the diffusion loss is used as the indicator of the training difficulty.**
>
> We would like to clarify that the diffusion loss defined in Section 4.5 is *not used to measure training difficulty*. Instead, it is *designed to measure how well the image is reconstructed through the encoder-generator pipeline*. Specifically, we use it as an indicator of the encoder's vision detail information extraction capability, measured **before and after** the encoder's finetuning.
>
> Formally, this metric is intended to compute the value of Eq. (4), i.e., how much information in image $\mathbf{x}$ is preserved after encoder $E$'s extraction. In practice, we estimate it using the diffusion loss introduced in Eq. (2), which is a lower bound of Eq. (4). For calculating the expectation term $\mathbb{E}_{\mathbf{x}, \mathbf{\epsilon}, t}$ in Eq. (2), random sampling noises $\epsilon$ and timesteps $t$ is needed ($\mathbf{x}$ are test set images) in practice. Since random sampling involves result **comparableness**, to make the two results (before and after finetuning) comparable, we pre-sample and save two sets of random noises and timesteps, and re-use the same sets for both of the two evaluations.
>
> The pre-saving strategy makes the metric analogous to a "validation loss": while it is computed using the same loss function as training (Eq. (2)), it uses fixed randomness and test images. In contrast, the training loss cannot be used as the indicator, as the sampled $\epsilon$ and $t$ vary during different training iterations, making the results not comparable.
>
> Some of the above explanation is presented in Section 4.5. In the revised paper, we will add a section in appendix to give a more detailed introduction about how the metric is calculated and its role.

---

### Official Review · Reviewer_dYip · 2025-07-02

**Clarity:** 3
**Significance:** 3
**Originality:** 3
**Rating:** 4
**Confidence:** 4

**Summary:**

This paper proposes an enhancement to CLIP's ability to capture visual details by utilizing gradient feedback from unCLIP. Unlike its closely related work, DIVA, this approach specifically addresses the issue of feature space misalignment between the input of the generator and the output of CLIP's image encoder. To implement this, the main technical modifications include replacing the text-to-image generator with unCLIP and removing the MLP projector. The effectiveness of the proposed method is demonstrated through various benchmarks that assess visual detail accuracy.

**Questions:**

please refer to the weakness part.

**Ethical Concerns:**

["NO or VERY MINOR ethics concerns only"]

**Final Justification:**

Most of my concerns have been addressed. I will keep my original rating.

**Limitations:**

The limitations are properly discussed.

**Paper Formatting Concerns:**

There are no formatting concerns.

**Quality:**

3

**Strengths And Weaknesses:**

### Strengths:

- The proposed method demonstrates strong effectiveness, which is thoroughly validated through comprehensive experimental evaluations.

- The paper's focus on addressing the misalignment between the input space of generators and the output space of the CLIP image encoder is reasonable.

- The qualitative improvements in visual detail capture, as clearly illustrated in Figure 4, provide compelling evidence for the benefits of the fine-tuned CLIP approach.

### Limitations:
- Novelty:
The technical contribution in comparison to DIVA is somewhat limited.

- Performance Trade-off:
The reported degradation in zero-shot classification performance (Tab.8, Appendix C) raises concerns. While the authors attribute this to the model's preference for fine-grained details, the superior performance of DIVA (a comparable approach) undermines this explanation. Further investigation into this performance discrepancy would strengthen the paper.

- Implementation:
Training a separate unCLIP generator for each CLIP image encoder creates significant computational overhead and limits implementation flexibility. Moreover, the effect of the scale of the training dataset on the generalizability of the CLIP image encoder during the unCLIP training process remains an unresolved question that deserves further discussion.

- Training Procedure:
A two-stage ablation—first training the MLP projector and then fine-tuning the image encoder while keeping the projector's weights fixed—could provide clearer insights into the role of the projector head. The current joint training strategies in the ablation may introduce unnecessary instability to the optimization process.

---

> ### Author Rebuttal · Authors · 2025-07-31
>
> We thank the reviewer for the thorough review and constructive comments. We are glad that the reviewer finds the paper's focus reasonable, the experimental evaluation comprehensive and effective, and the qualitative results compelling.
>
> Below we provide point-by-point responses to the reviewer's concerns. We are more than willing to provide additional clarification should any questions remain.
>
> > **1. Novelty: The technical contribution in comparison to DIVA is somewhat limited.**
>
> We acknowledge that our approach shares the high-level similarity with DIVA in using pretrained generative models to improve CLIP. The key contribution of our work lies in *dentifying and addressing the encoder-generator feature space mismatch issue* which is not well-addressed by prior work. Specifically, we *find* that an existing type of generative models (*unCLIP*) *offers a more seamless and effective CLIP enhancement framework*, and achieve better performance on various benchmarks.
>
> ---
> > **2. Further explanation of the performance degradation in zero-shot classification task.**
>
> As discussed in our response to Reviewer j8dh's last comment (where we use the visualization result of the open-vocabulary semantic segmentation task, i.e. Figure 2, to help analyze models' preferences), **our finetuned model corrects most of the foreground or other false predictions in background areas that the original CLIP model tends to rely on for classification**, resulting in performance degradation in the classification task. This is consistent with our previous attribution wroten in Appendix C (model's preference for fine-grained details).
>
> Regarding DIVA's performance, it can be seen in Figure 2 that **DIVA behaves similarly to the original CLIP in segmentation** (quantitative results in Table 2 also verify this), such as showing many foreground semantics incorrectly predicted in background areas (the rightmost giraffe example) or showing some other false predictions in background areas (the leftmost cow example), making the foreground semantics dominate the prediction results. This phenomenon can explain DIVA's maintained zero-shot classification performance compared to the original CLIP model shown in Table 8 in Appendix C.
>
> P.S. We may also observe that although DIVA shows marginal improvements on the pixel-level segmentation task in Table 2 and Figure 2, it achieves notable improvements on another visual-detail-related task MMVP-VLM in Table 1 (although falls behind our method). This may be because the MMVP-VLM benchmark emphasizes recognition of *visual patterns* (e.g., object orientation, state, relation, etc.), which are less related to the vision *class labels* used in segmentation and classification benchmarks. Therefore, the visualization of segmentation results can offer useful insight into classification performance of different models but are relatively *less* indicative for explaining performances on the MMVP-VLM benchmark.
>
> ---
> > **3.(1). Training a separate unCLIP generator for each CLIP image encoder creates significant computational overhead and limits implementation flexibility.**
>
> This is indeed a limitation of our approach which has been summarized in the last paragraph of Section 5. And we have found that (in Appendix B.1, and mentioned in Section 4.1) some existing pretrained CLIP models share similar embedding spaces thus they can share a same unCLIP model for reducing computational overheads, which is a trick solution for some models.
>
> In addition, training unCLIP generators actually does not take too many resources because there exist many open-sourced well-trained powerful generative models in the community such as Stable Diffusion. And in the codebase used in our experiments the unCLIP models are finetuned versions of Stable Diffusion, making the training overhead relatively affordable. For example, the unCLIP model conditioned on pretrained SigLIP embeddings used in the experiment is trained by ourselves, which took about 5 days on 8 A100-40G gpus. Such an experimental scale is much smaller than training large generative models from scratch, and can be implemented in a simple single-node experimental environment. These details are summarized in Appendix B.2.
>
> ---
> > **3.(2). Effect of the scale of the training dataset during ~~unCLIP~~ un$^2$CLIP training process.**
>
> (We assume the reviewer intended to refer to un$^2$CLIP rather than unCLIP, as the generative models like unCLIP or Stable Diffusion are usually adopted from existing checkpoints whose pretraining dataset scales are typically in large-scale such as LAION-5B to ensure general generation capability.)
>
> We thank the reviewer for raising this important question. The effect of the training dataset scale during the CLIP's generative finetuning process is indeed an unresolved question in this line of research. By default, we follow DIVA and use CC3M as the training dataset for fair comparison. We observe that the concurrent work GenHancer also adopts this setting. And there are no related ablation studies about the training dataset in all of these papers, making this issue an open question.
>
> **Experimental Setup.** Following the reviewer's suggestion, we study the effect of the training dataset scale within our un$^2$CLIP framework on the generalizability of the finetuned model. For this ablation experiment, we evaluate not only on the MMVP-VLM benchmark but also on open-vocabulary segmentation tasks to comprehensively assess generalization.
>
> **Training on ImageNet-1K.** We first switch from CC3M to another widely-used dataset ImageNet-1K, a class-balanced dataset with ~1.3M images across 1000 categories. We keep the number of training iterations the same as with CC3M to isolate the effect of dataset content. As shown in the table below (segmentation results are averaged over 8 datasets), the ImageNet-finetuned model achieves competitive results with the CC3M-finetuned one on MMVP-VLM. This may be because the MMVP-VLM dataset is on a small scale, whose image patterns can be well covered by both CC3M and ImageNet training sets. (Actually, part of the images in MMVP-VLM are taken from ImageNet, as introduced in MMVP-VLM's paper [9].)
>
> However, on segmentation benchmarks, interestingly, the ImageNet-finetuned CLIP cannot bring consistent performance improvement. Note that ImageNet-1K is a highly structured dataset (e.g., class-balanced, not containing the "person" class), which is different from web-collected datasets, such as CC3M, CLIP's training dataset WebImageText, and unCLIP's training dataset LAION-5B, in terms of data distribution. This raises doubts about **whether dataset content (distribution) or dataset scale results in poor generalizability**.
>
> **Training on 10% CC3M images.** To further investigate the above-mentioned question, we conducted another experiment training on a randomly sampled subset of CC3M, with the training sample numbers (~0.3M) even smaller than ImageNet-1K. As shown in the last row of the table, this model achieves better generalizability than the ImageNet-finetuned one, suggesting that **finetuning dataset distribution** (e.g., source diversity, similarity with CLIP's pretraining dataset) **plays a more crucial role** than finetuning dataset scale in achieving better generalized finetuned CLIP models.
>
> |Method|un$^2$CLIP Training Dataset Scale|MMVP-VLM|CLIP Seg.|MaskCLIP Seg.|SCLIP Seg.|ClearCLIP Seg.|
> |-|-|-|-|-|-|-|
> |Original CLIP|N/A|19.3|5.3|14.7|24.0|34.4|
> |un$^2$CLIP - CC3M (default)|~3M|**32.6**|**5.8**|**15.6**|**25.9**|**34.8**|
> |un$^2$CLIP - ImageNet-1K|~1.3M|**32.6**|5.3|13.9|24.0|33.0|
> |un$^2$CLIP - 10% CC3M|~0.3M|31.1|**5.8**|15.3|**25.9**|34.5|
>
> ---
> > **4. Two-stage training of the ablation experiment about the projector.**
>
> Thank you for this constructive suggestion. We agree that training the projector first then freezing it during the image encoder finetuning leads to a more stable optimization process compared to the joint training strategy used in our initial ablation. Following the reviewer’s advice, we have conducted this two-stage training experiment:
>
> An implementation detail is that the first stage's configuration adopts a larger learning rate (1e-5) than the second stage, because the number of trainable parameters is relatively smaller (only the projector). Similar configurations are also adopted in other multi-stage training frameworks such as LLaVA.
>
> The last row of the following table reports the result. We observe that it achieves a slightly suboptimal performance compared to our default setting. This is because projection layers are actually not needed in our framework, where the image encoder and generator are already aligned in unCLIP. In this 2-stage experiment, even though the projector was separately trained in advance, it may still be difficult for the projector to learn to achieve the previous encoder-generator aligned environment perfectly.
>
> |Method|MMVP-VLM|
> |:-|:-:|
> |Original CLIP|19.3|
> |un$^2$CLIP default (no projector)|**32.6**|
> |Jointly trained projector (random init.)|16.3|
> |Jointly trained projector (identity init.)|30.4|
> |2-stage training strategy|30.4|
>
> We will incorporate these added experiments into the revised paper to make the ablation studies more comprehensive.

---

### Official Review · Reviewer_j8dh · 2025-07-04

**Clarity:** 3
**Significance:** 3
**Originality:** 3
**Rating:** 4
**Confidence:** 4

**Summary:**

This paper proposes un²CLIP, a method to enhance CLIP’s ability to capture fine-grained visual details without requiring additional text-annotated data or retraining from scratch. By leveraging the unCLIP generative framework—where a diffusion-based image generator is conditioned on CLIP image embeddings—the authors invert the process: they finetune the CLIP image encoder using the fixed unCLIP generator so that the encoder’s embeddings retain more image information while remaining aligned with the original text encoder. The proposed approach is shown to yield significant improvements over both baseline CLIP and prior generative model-based enhancements (notably DIVA) on benchmarks such as MMVP-VLM, open-vocabulary semantic segmentation (dense prediction), and vision-centric multimodal LLM tasks.

**Questions:**

See the weaknesses. I generally think the method is simple and elegant, but the performance on some benchmarks hinders me from giving high scores.

**Ethical Concerns:**

["NO or VERY MINOR ethics concerns only"]

**Final Justification:**

My concerns have been addressed. Overall, the idea of this work is interesting, but the novelty seems a little limited. I lean to accept the paper, but I will not debate if it is rejected.

**Limitations:**

Yes.

**Paper Formatting Concerns:**

No.

**Quality:**

3

**Strengths And Weaknesses:**

Strengths:
- Though the idea of inverting unclip is simple, the method is well-motivated, and the experimental results are quite good. It can improve the performance of CLIP on multiple tasks and benchmarks. Compared to other competitors, the improvement is significant.
- The paper provides comprehensive experiments, including multiple baselines, detailed ablation studies, and both quantitative and qualitative results that strongly support its claims.
- The paper is generally well-structured and readable, with logical development from motivation, through theoretical grounding, to empirical validation.

Weaknesses:
- My main concern is that the improvement of un^2CLIP on MLLM evaluations is quite marginal. As shown in table 3, it performs similar to DIVA on MMVP and general benchmarks, which indicates the general image understanding ability of un^2CLIP remains the same. While MLLM related tasks are important, this is one of the most important metrics to evaluate the image encoder. Also, on traditional tasks like zero-shot classification and retrieval, its performance drops compared to original CLIP, showing that generative training focuses more on details while ignoring the global semantics.

---

> ### Author Rebuttal · Authors · 2025-07-31
>
> We thank the reviewer for the through review and the positive comments on the paper's motivation, writing, the elegance of the method, and the comprehensive experiments.
>
> Below, we provide point-by-point responses to the reviewer's concerns. We are more than willing to provide additional clarification should any questions remain.
>
> > **Table 3: Performances of un$^2$CLIP and DIVA on MMVP benchmark are similar.**
>
> This may be partially attributed to the relatively small scale of the MMVP benchmark, which includes only 150 sample pairs. A relatively smaller sample size may reduce the statistical significance of performance differences, especially in some competitive settings. In contrast, on other vision-centric benchmarks that contain more evaluation samples (e.g., CV-Bench 2D COCO that contains 800+ samples), un$^2$CLIP shows more consistent and significant improvements over prior methods.
>
> ---
> > **Table 3: Performances on general benchmarks are similar, which indicates the general image understanding ability of un$^2$CLIP remains the same.**
>
> **We agree with the reviewer’s understanding.** General MLLM benchmarks typically do not involve questions that require visual detail understanding ability. For example, in the POPE benchmark that evaluates object hallucination, questions are typically of the form "Is there a [something] in the image?". Such questions usually rely more on high-level semantics than visual details. And it can be seen in Table 3 that most CLIP visual detail improvement methods, including ours, perform similarly to the original CLIP on these benchmarks. **These general benchmarks are not the main focus of our work (and other related CLIP improvement methods) since we aim to improve CLIP's visual detail capturing ability.**
>
> On the other hand, the maintained performance to the original model on these general benchmarks *suggests that our finetuned CLIP model preserves the general, global semantics on MLLM tasks*, which is different to the conclusion drawn from the results of the traditional zero-shot classification task. We give some analysis in our response to the next comment below.
>
> ---
> > **On traditional tasks (zero-shot classification and retrieval), un$^2$CLIP's performance drops compared to original CLIP, showing that generative training focuses more on details while *ignoring* the global semantics.**
>
> We would like to point out that the global semantics are not **"ignored"** by the un$^2$CLIP-finetuned CLIP model. A more exact conclusion is that the global semantics are **not highlighted** in the un$^2$CLIP-finetuned CLIP model (i.e., global semantics **still exist**). Below are our analysis.
>
> First, as noted in our response to the previous comment, the results on MLLM general benchmarks suggest the finetuned CLIP have similar general image understanding ability with the original CLIP model. However, it is different from the observation of the results on the zero-shot classification task where the performance drops are relatively larger.
>
> The key difference of these two tasks is that the CLIP features fed to MLLM are processed by later learnable heads (projector and LLM in the system), while in zero-shot classification, CLIP features are directly used for the final prediction. The learnable heads in the former architecture offers a re-extracting opportunity for the required (global) semantics. While the learnable heads do no appear in the zero-shot classification framework. **This means that the frozen CLIP model should have highly aggregated global features to achieve better results on the training-free classification task, which does not exactly measure how the global semantics exist in the model.*
>
> To further illustrate, let us take an example from the visualization result of the open-vocabulary semantic segmentation experiment. In the rightest column of Fig. 2 in our paper (the giraffe example), it can be seen that the original model extracts more foreground semantic (giraffe) on the final output, while our finetuned model assigns more accurate semantics to the image. The behaviour of the original model, although misclassified many background areas, is obviously more easier to achieve success in the global-semantic classification task (the extreme case would be that it offers the global semantic label to every pixel in the image). On the other hand, it can be also seen that our finetuned model **does not "ignore"** the global semantic of the image. It just assigns the semantic to correct areas. Such a behavior makes it fall behind on the traditional classification tasks.

---

### Official Review · Reviewer_rnBc · 2025-07-05

**Clarity:** 1
**Significance:** 2
**Originality:** 1
**Rating:** 3
**Confidence:** 4

**Summary:**

This paper applies the unCLIP-inversion framework, which finetunes the image encoder of CLIP with a frozen generative module of unCLIP. Benefiting from the generator conditioned on the CLIP image embedding, the image encoder is able to capturing more rich visual details and improve the performance of MMVP-VLM, open-vocabulary segmentation and vision-centric MLLM tasks.

**Questions:**

please see weaknesses.
The method in this paper is reasonably designed and has superior performance compared with other CLIP methods that introduce generators. However, it is limited to comparisons with such methods and cannot strongly prove the promotion of the generative model to the image encoder of CLIP. Moreover, this paper also needs to prove that the generator will not undermine the initial image-text alignment ability of the CLIP model.

**Ethical Concerns:**

["NO or VERY MINOR ethics concerns only"]

**Final Justification:**

Thanks for the author's clarification. My concerns have been addressed.

**Limitations:**

see weaknesses

**Quality:**

2

**Strengths And Weaknesses:**

Strength
1. This work makes full use of the pre-alignment of CLIP image features and generative models in unCLIP, allowing the image encoder to benefit from the generative model in a relatively unified feature space.
2. The method in this paper only relies on pure image data and has a relatively low fine-tuning cost.

Weakness
1. Although this method freezes the text encoder of CLIP and claims to retain the initial image-text alignment capability of CLIP, the lack of comparison with the base CLIP model (such as SigLIP ViT-SO-14@384) in the experiments raises doubts about whether generative fine-tuning would undermine the initial discriminative ability of CLIP.
2. The comparative methods of the experiment only selected models that introduced the generator. Although this method outperforms them, to prove that the generator indeed brings more abundant visual details and knowledge to the image encoder, comparisons with other CLIP methods that do not use the generator are necessary.

---

> ### Author Rebuttal · Authors · 2025-07-31
>
> We thank the reviewer for the detailed review and for recognizing the reasonableness of our method and its superior performance compared to other generator-based approaches.
>
> We have carefully taken every comment and our responses are as below. We are more than willing to provide additional clarification should any questions remain. If you find that your concerns have been resolved, we would be grateful if you could consider updating your score.
>
> > **W1. Lack of comparison with the base CLIP model (such as SigLIP ViT-SO-14@384) in the experiments, which raises doubts about whether generative fine-tuning would undermine the initial discriminative ability of CLIP.**
>
> Actually, *in all of our experiments (Tables 1-3 in Section 4.2-4.4), our method have been compared with the base CLIP models.* The **"Original"** rows in Tables 1-3 represent the performance the base, original pretrained CLIP models. And it can be seen that our un$^2$CLIP finetuned models outperform the base CLIP models on these vision-detail-related tasks. For example, in the last group of Table 1, un$^2$CLIP outperforms the base CLIP model SigLIP ViT-SO-14@384 on the MMVP-VLM benchmark (41.5 vs 37.0). These results suggest that our generative finetuning does not undermine the discriminative ability of CLIP, but rather improves its capacity to capture fine-grained visual information.
>
> Furthermore, we checked our manuscript and find that this confusion may stem from that we did not highlight in the "Compared Methods" paragraph of Section 4.1 that the "Original" CLIP model is also one of the compared baselines in our experiments. We will add this clarification in that paragraph of the revised paper to make this point clearer.
>
> ---
> > **W2. & Q.(1). The comparative methods of the experiment only selected models that introduced the generator. Comparisons with other CLIP methods that do not use the generator are necessary.**
>
> We speculate that this concern may be related to the reviewer's previous **W1** question. As mentioned in our response to the previous question, our experiments include comparisons with the original base CLIP models, which do not involve the generator.
>
> *Regarding other methods,* to the best of our knowledge, existing works that focus on improving pretrained CLIP's visual detail capturing ability, at the upstream task-agnostic level, are all generative model based approaches, including the previous DIVA and the concurrent GenHancer. *These are directly compared with in our experiments, and our choice of baselines is consistent with these works.* If the reviewer could point out the some specific works that should be also compared with, we would be more than willing to discuss with and try our best to conduct further experiments to give empirical comparisons during the following discussion period.
>
> P.S. Taking W1 & W2 together, considering that the reviewer may also be referring to a comparison between our un$^2$CLIP (with the CLIP backbone) and the SigLIP model, below is our response from this aspect:
> Our goal is not to develop a stronger CLIP backbone like SigLIP, but to enrich existing pretrained CLIP's visual detail capture ability. SigLIP involves a different training objective and configuration compared to the CLIP, making it not directly comparable in this context. Such comparisons are also not adopted in prior or concurrent works in this field (DIVA and GenHancer). That said, our method (and related work DIVA and GenHancer) are orthogonal to and can be combined with different CLIP backbone improvement works such as SigLIP. As shown in Table 1 of our paper and mentioned in our response before, our method can improve such backbones on vision-detail-related tasks.
>
> ---
> > **Q.(2). The paper needs to prove that the generator will not undermine the initial image-text alignment ability of the CLIP model.**
>
> When developing our approach, we have taken the image-text alignment ability into consideration. Specifically, since the input space of the unCLIP generator lies within the CLIP initial image-text embedding space, for reducing the potential language-shift when finetuning the image encoder, we freeze the parameters of the unCLIP generator to encourage the optimized image embeddings staying close to unCLIP's original input space. These are introduced in Section 3.2.
>
> Moreover, *the experimental results in Section 4.2 (MMVP-VLM evaluation) and 4.3 (open-vocabulary segmentation) implicitly validated the preserved image-text alignment property*: These two tasks both use the original CLIP text encoder together with the finetuned image encoder to obtain the prediction results, meaning that the alignment should be maintained for successful performance.
>
> For instance, the segmentation results in Section 4.3 are obtained by comparing the local patches of image features to the candidate text features. The results on various datasets (COCO, ADE, etc.), containing various classes including both indoor and outdoor scenarios, show the improved performance when using our finetuned model. On the contrary, if the finetuned model breaks the image-text alignment ability, it would be difficult to achieve consistent improvement over different datasets.

---

> > ### Author Response · Authors · 2025-08-05
> > **Kindly Request for Your Feedback!**
> >
> > Dear Reviewer rnBc,
> >
> > We appreciate your time and effort in reviewing our paper. We have carefully responded to all the concerns raised in your review, including:
> >
> > 1. **Comparison with base CLIP models:** which we clarified is already included throughout Tables 1–3 (labeled as “Original”).
> >
> > 2. **Comparison with non-generative baselines:** where we explained that, to the best of our knowledge, prior and concurrent works in this direction (e.g., DIVA, GenHancer) are all generator-based, and our selection of baselines is consistent with theirs.
> >
> > 3. **Preservation of image-text alignment:** which we supported with both framework design choices and experimental verifications (Sections 4.2-4.3), and discussed in detail in the above rebuttal.
> >
> > We hope these responses fully address your questions. We are more than happy to provide further clarifications if needed. If these clarifications resolve your concerns, we would be very grateful if you could reconsider your rating.
> >
> > Thank you again for your time and feedback.
> >
> > Sincerely,
> > The authors of Paper 21571

---

### Decision · Program_Chairs · 2025-09-17

**Decision:**

Accept (poster)

**Comment:**

The paper proposes a method that improves the fine-grained recognition abilities of CLIP via an  encoding-decoding training approach.

The paper received four reviews with mixed, mainly borderline ratings: BR (no answer after rebuttal) - A - BA - BA

The reviewers mainly highlighted that the proposed inversion module to enhance the compositional understanding capabilities of CLIP-like models is an interesting and intuitive approach, and that the proposed algorithm should be easy to integrate into CLIP architectures.
Most weaknesses raised by reviewers have been addressed in the rebuttal.

The AC follows the majority of the reviewer voting and recommends accepting the paper.
The AC would encourage the authors to integrate the findings of the rebuttal in the CR version of the paper.